# Has resistance to chlorhexidine increased among clinically-relevant bacteria? A systematic review of time course and subpopulation data

Stephen Buxser *

Select Bio Consult, LLC, Indianapolis, Indiana, United States of America

* sbuxser@selectbioconsult.com

**Data Availability Statement:** All relevant data are within the paper and its Supporting Information files.

**Funding:** This study was funded under a contract between Select Bio Consult, LLC and Molnlycke

## Abstract

Chlorhexidine (CHX) was introduced for use as an antimicrobial more than 70 years ago. CHX has been and continues to be used broadly for disinfecting surfaces in medical and food service facilities as well as directly on skin of humans and animals. Considering its widespread use over many decades, questions of resistance to CHX have been raised. Additionally, questions of possible coincident resistance to the biocide and resistance to clinically relevant antibiotics have also been raised. A number of important questions remain, including is there consistent evidence of resistance, what is the degree of resistance, especially among clinically isolated microbial strains, and what is the degree of resistance compared to the typical concentrations of the biocide used? Data for microbial species isolated over the last 70+ years were compiled to construct as complete a picture as practical regarding possible resistance, especially among species in which resistance to commonly used antibiotics has been noted to be increasing. This is a compilation and analysis of individual MIC values for CHX reported in the literature, not a compilation of the conclusions individual authors reached. The data were analyzed using straight-forward and robust statistical procedures to detect changes in susceptibility to CHX over time, i.e. linear regression. Linear regression was supplemented with the use of nonlinear least squares regression analysis to detect the presence of population parameters associated with subpopulations of microbial strains which exhibit increased resistance to CHX. *Pseudomonas aeruginosa*, *Klebsiella pneumoniae*, and *Acinetobacter baumannii* were all found to have an increased resistance to CHX over time with the most profound change detected in *A. baumannii*. Additionally, subpopulations with log-normal distributions were found consistent with the presence of a baseline subpopulation of susceptible strains and a subpopulation with increased resistance to CHX. However, the CHX-resistant subpopulations did not correlate exactly with antibiotic resistance, so details of the relationship remain to be addressed. Increased resistance over time was not detected for *Escherichia coli*, *Enterobacter faecalis*, *Staphylococcus aureus*, or *Candida albicans*, although a subpopulation with greater than baseline resistance to CHX was detected among strains of *E. faecalis* and *C. albicans*. A difference in susceptibility to CHX was also detected between methicillin-resistant (MRSA) and methicillin-sensitive (MSSA) *S. aureus* strains. The levels of resistance to CHX detected were all markedly lower than concentrations routinely used in medical and food service

Health Care, Surgical Marketing, in Peachtree, GA. The url is: https://www.molnlycke.us/contact-us/. The contact person at Molnlycke is Jason Liles. Molnlycke is the manufacturer and marketer of products including formulations containing chlorhexidine. SB was contracted as an independent consultant employed by Select Bio Consult, LLC and has no other affiliation with Molnlycke or any of its affiliates. The funder provided support in the form of salaries for the author [SB], but did not have any additional role in the study design, data collection and analysis, decision to publish, or preparation of the manuscript. The specific roles of these authors are articulated in the author contributions section. The funders had no role in study design, data collection and analysis, decision to publish, or preparation of the manuscript. Select Bio Consult, LLC is incorporated in Indiana and its sole owner and employee is Stephen Buxser, Ph.D. There are no additional relationships financial or non-financial between Select Bio Consult, LLC, Stephen Buxser, and Molnlycke Health Care.

**Competing interests:** I have read the journal's policy and the authors of this manuscript have the following competing interests: SB (Select Bio Consult, LLC) was funded as an independent contractor by Molnlycke Health Care, USA. Independence stipulated that Molnlycke had no role in study design, data collection and analysis, decision to publish, or preparation of the manuscript. The author was a temporary paid consultant for the project described in the manuscript but has no other financial or non-financial relationship with Molnlycke. The term of the consultancy strictly covered the term to complete the relevant research and prepare the results for publication and does not include any other financial transactions before, during or since. This includes no ownership, employment, board membership, patent applications, research or travel grants, gifts or honoraria. There is no other competing interest between Molnlycke and Select Bio Consult or its sole owner and employee, Stephen Buxser A Ph.D. The contract between Select Bio Consult, LLC [SB] and Molnlycke Health Care does not alter adherence to PLOS ONE policies on sharing data and materials.

applications. Reaching conclusions regarding the relationship between antibiotic and CHX resistance was complicated by the limited overlap between tests of CHX and antibiotic resistance for several species. The results compiled here may serve as a foundation for monitoring changes in resistance to CHX and possible relationships between the use of CHX and resistance to antibiotics commonly used in clinical medicine.

## Introduction

The antimicrobial properties of chlorhexidine (CHX) were described in 1954 [1]. Since then the compound has been used as a topical antiseptic incorporated into many commercial products, especially since the 1970s. For a list of products and marketing dates see https://go. drugbank.com/drugs/DB00878. Further history of CHX use is found in https://www. chlorhexidinefacts.com/history-of-chlorhexidine.html. CHX is commonly used orally in dentistry for treatment of inflammatory dental conditions as well as for skin disinfection before surgery, for sterilization of surgical instruments, and as a cleaning and disinfecting agent for surfaces in food preparation and medical facilities. CHX is a broad spectrum disinfectant with activity against Gram-positive and Gram-negative bacteria, fungi and viruses [2]. Recently, CHX has been considered in the context of resistance to antimicrobial agents [3, 4]. Although CHX is not an antibiotic used in clinical medicine, the possibility has been raised that the use of CHX can result in resistance to itself and to conventional antibiotics used in human and animal medicine. Shared resistance mechanisms may include decreased permeability and increased efflux transporter activity, which may be consistent with biocide resistance imparting antibiotic resistance [5]. Although it has been demonstrated that biocide resistance can be induced in many infectious bacteria and that some antibiotic resistant bacteria also have been reported to be resistant to one or more biocides, a direct causal relationship between appearance of biocide resistance and antibiotic resistance has not been definitively demonstrated. How much resistance to biocides is present in bacteria isolated from either the environment or in clinical settings has not been extensively explored. The work described below is an attempt to address the fundamental question, is there strong evidence of increasing resistance to biocides, in particular resistance to CHX, among important infectious bacteria and fungi?

The focus is purposely narrow. The approach was to compile data reporting CHX antimicrobial activity from peer-reviewed literature for microbial strains isolated over as long a period of time and covering as many important infectious organisms as there is sufficient published information to draw inferences or conclusions. The time course of measurements for CHX susceptibility and resistance varies from species to species but is at least 20 years, and data for strains isolated over more than 50 years have been included in this report. Since resistance to antibiotics varies among microbial species, an important question is whether species-to-species differences in resistance to CHX has occurred. Additionally, within a species exhibiting CHX resistance, does resistance appear as a small increase in resistance among many strains or as a high degree of resistance among a few strains? Considering the broad use of CHX, the absence of CHX resistance for some species is as important to document as is the presence and degree of CHX resistance in other species. Such differences are important to consider so that CHX resistance among resistant bacterial species may be monitored more aggressively than others.

The methodology was to compile and analyze antimicrobial values for individual strains and not to compile conclusions, statistics or summaries from previous studies. Thus, the review is accurately described as a systematic review but, because the goal was to compile raw data, the analysis is not meta-analysis. The information compiled for important bacterial

infections provides insight into changes in susceptibility or resistance to CHX since the compound was introduced. The goal of this report is to provide an understanding of the degree to which CHX resistance has occurred over time in bacteria and fungi with emphasis on species with direct clinical relevance, including examining the relationship between general antimicrobial activity of CHX and whether this may affect resistance to clinically relevant antibiotics.

## Methods

In microbiology, the most typical scheme for evaluating antimicrobial potency is the use of two-fold serial dilutions. In the descriptions that follow, it is conceptually and statistically useful to use the term $\log_2$(MIC) which, as the term indicates, is the base two logarithm of the MIC value. The use of this form of MIC simplifies expression of MICs to a series of integers that are easily translated back to the usual MIC concentration where, for example, 4, 2, 1, and 0.5 μg/ml is converted to values of 2, 1, 0, and -1 in $\log_2$(MIC) units.

It is also important to distinguish the use of the words "mean" and "standard deviation" into their context sensitive meanings. When referring to a series of observations, mean and standard deviation refer to summary statistics of that restricted set of values, but in the context of populations in the descriptions below, mean and standard deviation refer to population parameters summarizing the properties of log-normal distributions.

### Protocol and registration

No review protocol exists for this report.

### Eligibility criteria

The sole criterion for eligibility for inclusion in the compiled data was a numerical value for minimum inhibitory concentration using chlorhexidine as an antimicrobial in an in vitro assay. Predominantly, the chlorhexidine form was chlorhexidine gluconate, although a few studies used the chlorhexidine diacetate form. If CHX was included in a formulation with alcohol, detergent, or any other known co-inhibitory molecule of microbial growth, the results were excluded to prevent interference by a synergistic effect. Most reports included results from testing of strains from medical facilities, especially hospitals, although initial findings included stains isolated from non-clinical settings. When included in final results, stains from non-clinical origins are specified as such. Otherwise, results included were from clinical samples from humans. Some studies specified anatomic locations of strains reported, but results were combined from all strains regardless of anatomic source.

### Information sources

Literature citations were accumulated over a period of 9 years beginning with a search of PubMed, Science Direct, and Cochrane Library databases in 2012 covering publications from as early as the 1970s up to the search date. Additional searches were carried out in March 2018 emphasizing publications appearing between 2012 and 2018 as found in Scopus, Biological Abstracts, International Pharmaceutical Abstracts, the Life Sciences Collection, and Medline. A final search for additional publications appearing between 2018 and August 13, 2020 was carried out in Science Direct and Dimensions (https://www.dimensions.ai/#). The PRISA flow diagram (Fig 1) summarizes the results and selections made for inclusion. Dates applied for MIC results from reference strains, predominantly ATCC strains, were the isolation dates stated in the source strain collection. Many reference strains were isolated in the 1940s providing the earliest isolation dates for strains compiled for this study.

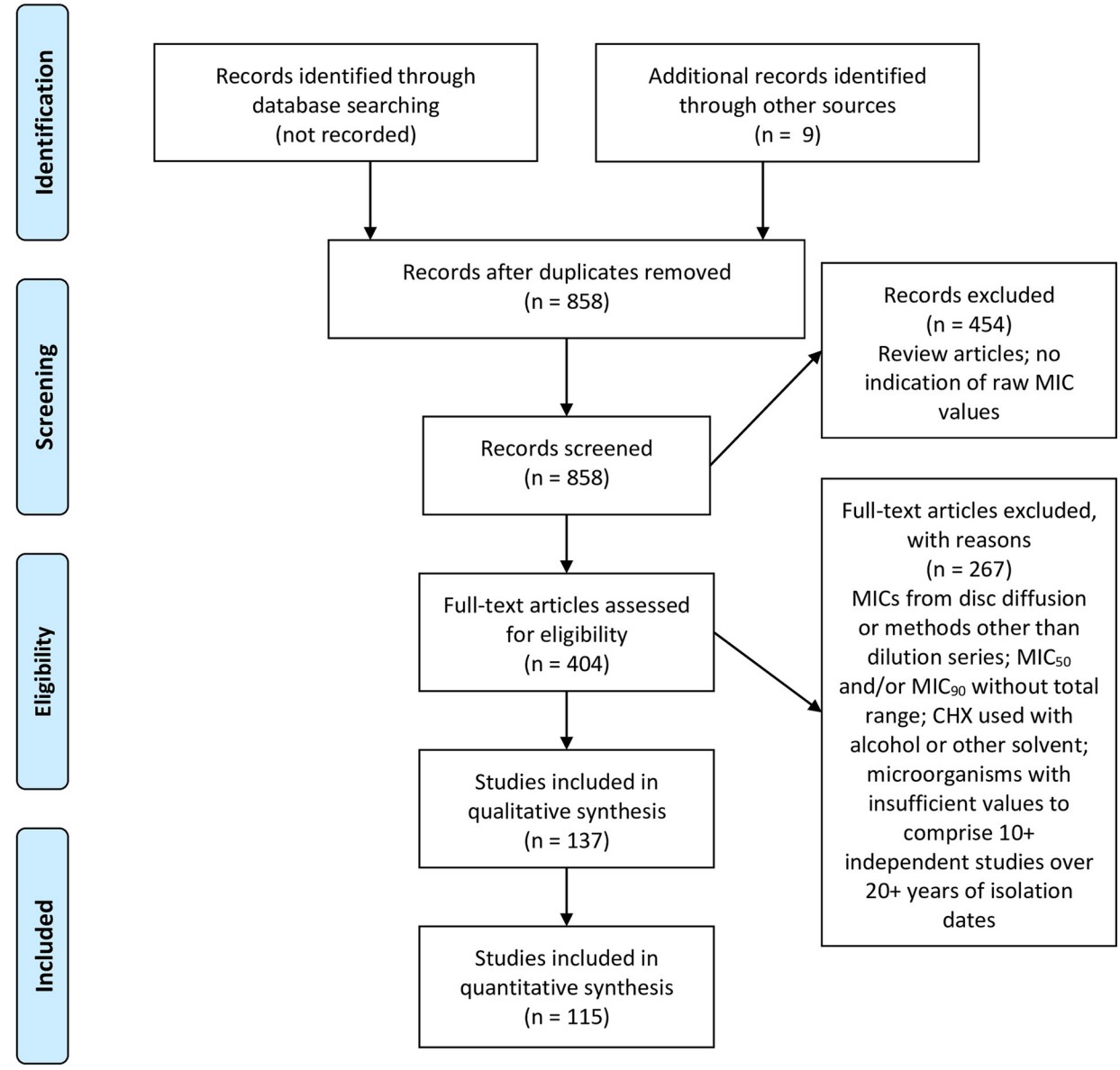

**Fig 1. Literature selection process.**

## Search

The search algorithm used for all searches was:

(chlorhexidine OR CHX) AND (resistance OR MIC OR "minimum inhibitory")

Cross-checking citation lists within publications found, including review articles, was used to extend finding relevant citations.

## Study selection

Other than checking for citations absent from other search results, review articles were excluded from consideration. Only reports that mentioned CHX and MICs in abstracts or titles were included for detailed consideration. Only results clearly stated as MIC values based

on dilutions, i.e. not including disc diffusion results, were compiled for use in detailed statistical analyses. Initially, results from any bacterial or fungal species were considered, but final analyses were completed only for the species with results analyzed below wherein a minimum of 10 independent studies were assessed for any species with isolation dates extending at least over a 20 year interval with typical results extending over 50 or more years of strain isolation dates.

### Data collection process & data items

Citations were selected for further review, if the titles or abstracts suggested that individual strains had been assessed for susceptibility to CHX using MICs. If so, full-text articles were obtained and screened for the presence of MIC values reported using in vitro serial dilutions. Most results were presented as MICs for individual strains, although many papers compiled information for more than one microbial species. Data were also included for values presented by authors as $MIC_{50}$ and/or $MIC_{90}$ values, if the number of strains with particular MIC values or the number of strains with MIC values within a narrow range of values could be determined. When necessary, randomization within a narrow range of possible MIC values was used to assign specific values to individual strains. Permutation runs were used to calculate the range of fitted parameters resulting from the randomizations. See S1 Appendix for details. Additionally, a date or date range for isolation of particular strains was required for inclusion. Typically, date ranges were over a two-year or shorter period. As detailed in S1 Appendix, randomization of dates was carried out within the range of isolation dates stated by authors to observe effects of isolation date uncertainties on statistical parameters calculated to detect presence or absence of changes in CHX susceptibility over time.

Randomizations of possible MIC values and isolation dates were used to assign specific values to all strains used in analyses. To account for uncertainty in MICs or isolation dates, a series of randomizations were carried out–termed "permutation runs"–to generate a series of data sets for each species. Time course datasets contained randomizations of both isolation dates and MIC values within the confines of the uncertainty in individual values. Parameters from the collection of permutation run datasets were used to assess the effect of uncertainty in the data on the time course parameters, especially on the slope of linear regression of MICs over time. At least 50 permutation runs were compiled for time-course evaluations for each species reported. Statistics of the fits of linear regression were tested using the F-test for linear regression, i.e. a test of whether the slope coefficient is different than zero. Linear regression parameters from the pooled permutation runs also included calculation of the standard error of the slope and the coefficient of determination ($r^2$), which are also useful for assessing the precision of linear regression parameters. p-values for F-tests and for $r^2$ values were compiled to observe variability among permutation runs. Linear regression analysis was run using Excel 2013 programs (LINEST).

Non-linear least squares regression was used to determine population and subpopulation parameters for the collection of strains from a single microbial species reported in the compiled MIC data regardless of date. Compiled MIC values for single microbial species–independent of collection date–were analyzed for conformance to a single or to the sum of two log-normal distributions. The presence of a single population indicated that no subpopulation of resistant strains was detected. Improved fit of a model for the sum of two log-normal distributions indicated the presence of a subpopulation with a base-line susceptibility to CHX and a second subpopulation with lower susceptibility/higher resistance to CHX. A parameter indicating the number of strains with lower susceptibility in the model with the sum of two log-normal distributions was used to estimate the size of the more resistant subpopulation. Non-

linear least squares regression was carried out using Solver in the Analysis Tool-Pak in Excel 2013. The software used for non-linear least squares regression was the generalized reduced gradient (GRG) nonlinear algorithm [6] incorporated in Microsoft (R) Excel 2013. Degree of improvement in the fit to the sum of two log-normal distributions was tested using both the F-test and the Akaike Information Criterion (AIC). Details are provided in S1 Appendix for non-linear least squares regression, the log-normal models tested, and the statistics used to assess quality of fits. At least 10 permutation runs were carried out for each non-linear least squares analysis. Typically, parameters derived from the permutation runs differed by a few percent among runs, and in all cases standard deviations of compiled parameter estimates were less than ten percent.

## Risk of bias in individual studies & across studies

Individual studies may exhibit bias, since studies of antimicrobial resistance are often initiated because a problem is suspected or has been demonstrated by other means. That is, a hospital concerned that they are experiencing problems with onsite antimicrobial resistance are more likely to test for resistance. If no problem is suspected, there is limited inclination to launch such a study. This could bias outcomes by resulting in detection of more resistance than would be detected if studies were more evenly distributed to include facilities where no problem is suspected. The degree of bias this may introduce is extremely difficult to assess, although the increasing number of investigations infers indirectly that resistance is increasing with time.

## Summary measures

Since this is not meta-analysis, summary measures such as risk-ratios or other measures based on summaries of data generated from individual reports were not used. Instead, individual MIC values were compiled across studies with MIC values for each particular species among the eight species included in the final analysis. Summary measures were derived by species to determine the time course of changes in CHX resistance using standard linear regression analysis and for detection of subpopulations within a species with differences in CHX susceptibility using non-linear least squares regression analysis, as described above and in S1 Appendix. Weighting in such models is dependent strictly on the number of MIC values included in the analysis. Thus, studies reporting many individual MIC values were weighted proportional to the number of MIC values for particular microbial species. When possible, analysis of means for subpopulations determined using non-linear least squares analysis were compared to CHX MICs for strains known to be resistant to clinically-relevant antibiotics. In several cases, as shown in detail below, strains with resistance to antibiotics were also resistant to CHX, although resistance to CHX was of low magnitude. It is important to keep in mind that such a result is correlative and not necessarily a demonstration of CHX causing antibiotic resistance.

## Synthesis of results

No meta-analyses were used.

## Additional analyses

The appearance of resistant strains are unlikely to occur en masse in any particular study but rather as a limited number of strains among a collection of non-resistant, strains. Therefore, techniques that attempt to reduce variability in a particular study, as is common in meta-analysis such as meta-regression, are inappropriate for analyses presented below. Such pooled

variance-reducing analyses are likely to reduce or completely disguise detection of resistant strains in the mass of data collected for a particular species across numerous study reports.

## Results

### *Ps. aeruginosa*

One of the most important organisms with increasing resistance to clinical antibiotics is *Pseudomonas aeruginosa*. Thus, determining resistance to antiseptics, including CHX, is unusually important. Simple linear regression of MIC values using CHX susceptibilities of strains isolated over an 80 year time interval showed a slight increase in slope of the line (Fig 2).

The results indicate a significant slope to the regression line corresponding to an increase in MIC values for CHX of 0.019 $\log_2$(MIC) units per year or approximately a 2-fold change for strains isolated over 50 years. The mean slope (± standard error of the mean, SEM) for linear regression was 0.019 (±0.0005) $\log_2$(MIC)/year using randomization of isolation dates and randomization of ~15% of the $\log_2$(MIC) values. The robustness of the linear regression is indicated by p-values associated with the F-tests of regression. The distribution of p-values,

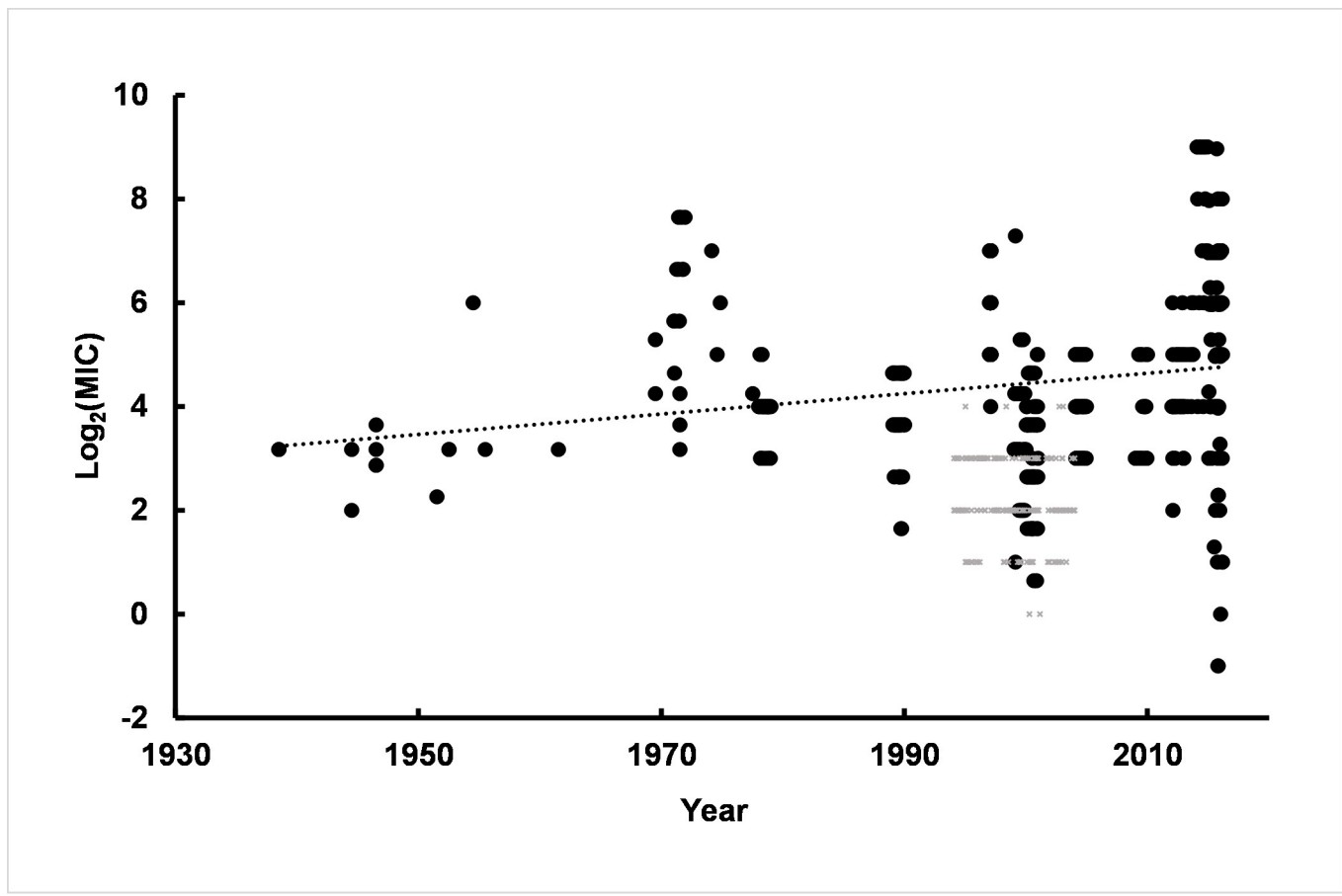

**Fig 2. Linear regression of $\log_2$(MIC) values over time for *Ps. aeruginosa*.** Values used in linear regression were from human clinical samples collected from approximately 1940 to 2016 (open circles). The dotted line is the linear regression line derived from the human clinical isolates. Small grey symbols are $\log_2$(MIC) values from non-clinical strains or strains isolated from animals and were not included in the linear regression shown. There are 523 $\log_2$(MIC) values compiled from 21 separate reports used in linear regression.

regression parameters and results of statistical tests are shown in S1 Fig and S1 Table in S2 Appendix.

Was the increase in resistance over time the result of a large number of strains with a low level of resistance or is there a small subpopulation of highly resistant strains? Additionally, was there a continuum of strains with gradually increased CHX resistance or were there discrete subpopulations with definable population parameters among the strains? To determine the distribution of resistance among *Ps. aeruginosa* strains, distributions were fitted to the $\log_2$(MIC) values (Fig 2), i.e. fitting log-normal distributions. The fit to the sum of two log-normal distributions was markedly better than the fit to a single log-normal indicating the presence of two subpopulations. The mean of the less-resistant subpopulation was 3.85 $\log_2$(MIC) units (~ 14 μg/ml). The second subpopulation indicated greater resistance to CHX with a mean of 4.81 (~ 28 μg/ml). Thus, two subpopulations of *Ps. aeruginosa* were detected including a subpopulation approximately 2-fold more resistant to CHX than the base subpopulation, which is consistent with values calculated from the linear regression time course. Non-linear least squares regression also provided an estimate of the relative number of strains associated with each subpopulation. The *Ps. aeruginosa* subpopulation associated with the higher mean value was approximately 62% of the 523 strains compiled for this report indicative of a large number of strains with a low level of increased CHX resistance. Detailed results of log-normal non-linear regression analyses are given in S2 Table in S2 Appendix.

Values for strains isolated prior to 1990 were pooled and analyzed. The pre-1990 results showed a single population parameter with mean value = 4.0 and a population standard deviation parameter = 0.88. The values are in reasonable agreement with the more susceptible mean and standard deviation parameters from the subpopulation of CHX $\log_2$(MIC) values over all dates shown in Fig 3 (mean = 3.85, standard deviation = 0.69). The small standard deviation indicated that the range of values over the population of strains isolated before 1990 was narrow (S3 Table in S2 Appendix). $\log_2$(MIC) values for strains isolated after 1990 were pooled and analyzed separately. The post-1990 subpopulation was consistent with the sum of two log-normal distributions with population parameters in close agreement with the values from the analysis of all strains regardless of isolation date (S2 Fig and S4 Table in S2 Appendix). However, the post-1990 subpopulation showed that approximately 3/4s of the strains were consistent with the more resistant subpopulation of strains. Additionally, the standard deviation of the population was broad compared to the pre-1990 strains suggesting the presence of strains with a broader range of levels of resistance compared to the pre-1990 population.

In summary, the pre-1990 strains are consistent with a single log-normal population, but the post-1990 population has both a subpopulation consistent with the pre-1990 strains and a second more resistant group of strains, approximately 76% of the total, with an increased resistance to CHX, albeit only an approximately 2-fold higher mean MIC. Summary parameter values associated with sub-populations are presented in Table 1.

Antibiotic resistance was reported by authors for 44 (8.4%) of the 523 strains evaluated from all isolation dates. Since 10 of the $\log_2$(MIC) values for the 44 antibiotic resistant strains were presented as $MIC_{50}$s, 20 permutation runs were carried out to generate values in the range designated by authors [7]. The mean of the 20 permutation runs was 5.74 $\log_2$(MIC) units (SEM = 0.020) and the standard deviation of the 20 permutation runs was 2.41 $\log_2$(MIC) units (SEM = 0.011). Thus, the values for the antibiotic-resistant strains are most consistent with the values for the more resistant subpopulation of CHX strains isolated after 1990. Unfortunately, the number of values for antibiotic-resistant strains was not sufficient to support conclusive non-linear least squares regression analysis. Nevertheless, resistance to conventional antibiotics appeared to be associated with *Ps. aeruginosa* strains that were more resistant to CHX than were strains isolated before 1990.

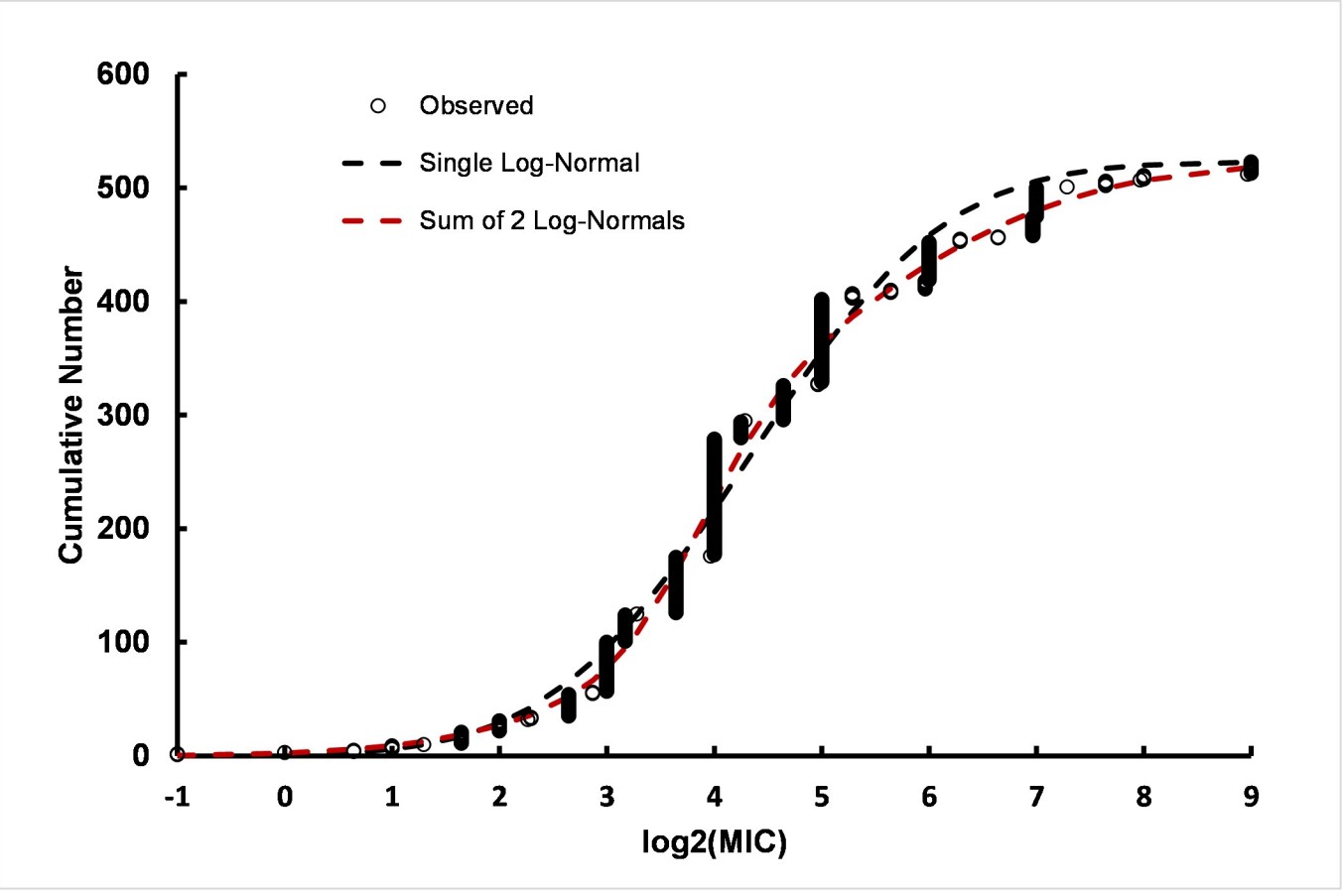

**Fig 3. *Ps. aeruginosa* with single log-normal (black dashed line) and sum of two log-normal distributions (red dashed line) fit to log$_2$(MIC)s for 523 strains for all dates.** The summary of ten permutation runs is provided in S2 Table in S2 Appendix.

### *K. pneumoniae*

Another extremely important organism to characterize with respect to its CHX resistance is *K. pneumoniae*. The time course of susceptibility to CHX is shown in Fig 4. The slope of the linear regression line, 0.0300 ± 0.001 log$_2$(MIC) units per year, indicated an increase in resistance of approximately 2.8-fold over 50 years.

**Table 1. Population parameters for CHX susceptibility of *Ps. aeruginosa*.**

|  | Total population (all dates) | Pre-1990 | Post-1990 |
|---|---|---|---|
| **Mean 1** | 3.85 | 3.99 | 3.84 |
| **Std Dev 1** | 0.69 | 0.88 | 0.57 |
| **Mean 2** | 4.81 | - | 4.77 |
| **Std Dev 2** | 1.97 | - | 1.95 |
| **n2** | 322 (61.5%) [A] | 124 (23.5%) [A] | 301 (75.7%) [B] |

All values are in log$_2$(MIC) units.

[A] n is the number (%) of values among the 523 associated with Mean 2 and Std Dev 2, all isolation dates.

[B] n is the number (%) of values among the 400 associated with Mean 2 and Std Dev 2, post-1990.

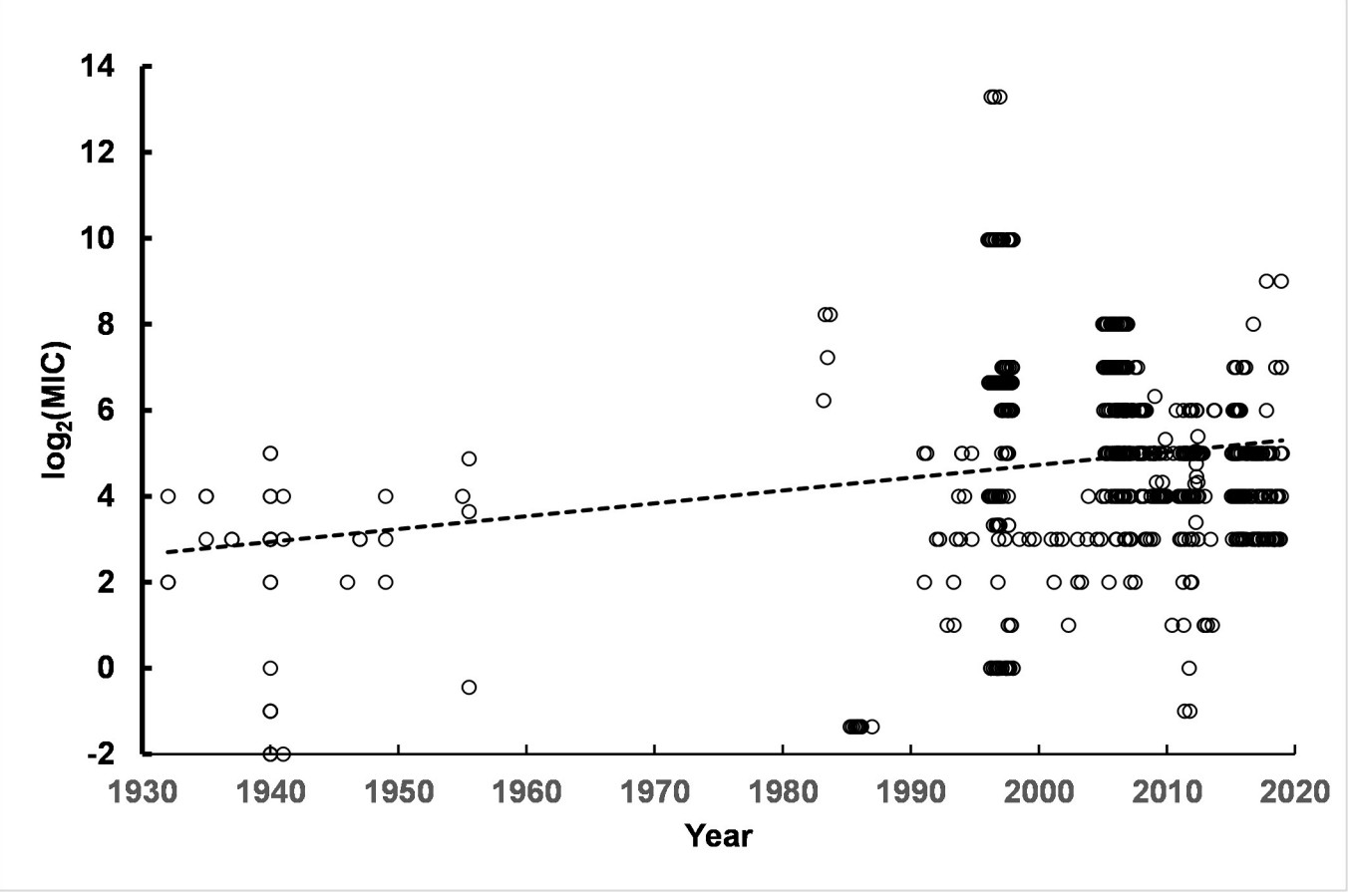

**Fig 4. Time course of *K. pneumoniae* susceptibility to CHX.** The results are from a single representative linear regression analysis from among 50 permutation runs. Details of linear regression parameters are given in S5 Table in S2 Appendix. Replicate permutations sampling within the range of uncertainties for isolation dates and MIC values as reported in the literature for the 21 studies compiled for this report indicated a small slope consistent with a small degree of decrease in susceptibility to CHX over the approximately 90 year time interval of isolation dates for *K. pneumoniae* strains.

The fit of a single or the sum of two log-normal distributions to the data (n = 714) indicated the presence of two subpopulations in the pooled data (S6 Table in S2 Appendix). One subpopulation had population parameters with mean = 3.90 log$_2$(MIC) (population standard deviation = 0.54 log$_2$(MIC) units). A more resistant subpopulation had population parameters with mean = 5.02 log$_2$(MIC) units (population standard deviation = 1.98 log$_2$(MIC) units). The log$_2$(MIC) values correspond to 12 µg/ml and 32 µg/ml, respectively, i.e. 2.1-fold difference in resistance to CHX consistent with the degree of change noted from the linear regression time course. Results from a representative permutation run are shown in Fig 5. As was observed for *Ps. aeruginosa* strains, the population standard deviation of the more resistant subpopulation was also markedly broader than the population standard deviation for the baseline *K. pneumoniae* strains.

Some of the reports identified strains with antibiotic resistance. In particular extended-spectrum beta-lactamase-resistant (EBSL) strains [8, 9], NDM-1 containing strains [10], carbapenem-resistant strains with sequence type ST258 [11], ST395 sequence type strains [12], ceftazidime-resistant strains [13], strains designated as ST147 [12], strains listed as carbapenem-resistant *K. pneumoniae* (CRKP) [9], and multi-drug resistant (MDR) strains [14] were reported with their corresponding MICs for CHX. A total of 205 CHX MIC values were listed

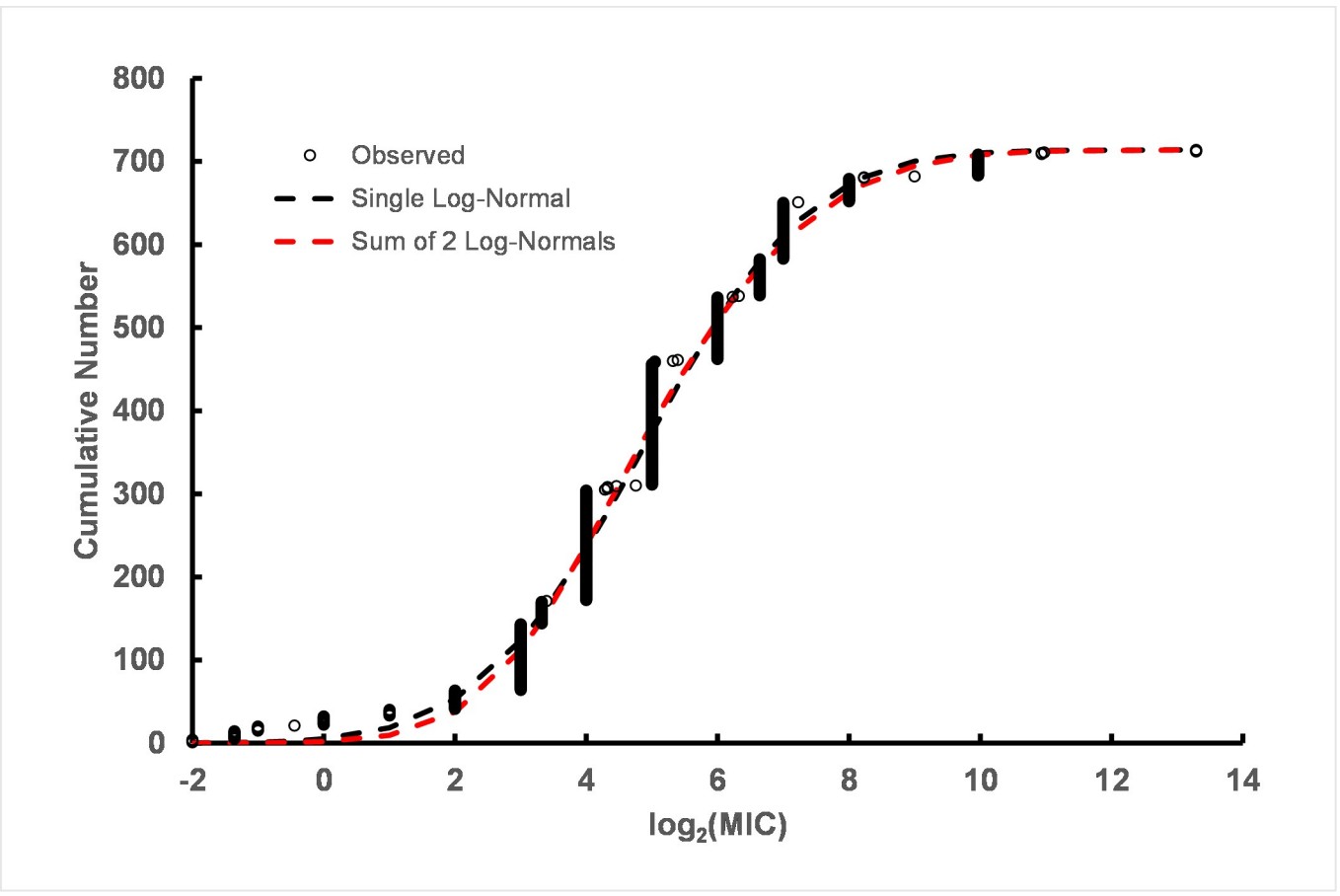

**Fig 5. Comparison of fits by single and sum of two log-normal cumulative distribution functions to pooled _K. pneumoniae_ data.** The improved fit of the sum of two log-normal distributions compared to a single log-normal was indicated by differences in AIC ranging from 11 to 68 and p-values for the F-test ranging from $1 \times 10^{-5}$ to $1 \times 10^{-3}$. See S6 Table in S2 Appendix for details.

for antibiotic-resistant strains among the studies included in this report. Pooled MICs found the mean (± Std Dev) for all of the antibiotic-resistant strains $\log_2$(MIC) values = 5.6 (± 1.6). The isolation years for the antibiotic-resistant strains ranged from 2006 to 2019, i.e. late in the time series presented in Fig 4. The mean (± Std Dev) for the strains not listed as antibiotic resistant, which were isolated in years from 1932 to 2019, was 4.6 (± 2.4). Permutation runs were used to generate data sets accounting for the uncertainty in 38 of 205 MICs for which $MIC_{50}$s and $MIC_{90}$s were reported in antibiotic resistant strains and 137 of 509 strains not listed as antibiotic-resistant. Student's t-test with unequal variances (Welch's test) found p-values ranging from $5 \times 10^{-8}$ to $3 \times 10^{-9}$ consistent with distinguishable differences in MICs between the two groups.

The permutation runs found very robust population parameters for antibiotic-resistant _K. pneumoniae_ strains, as given in detail in S7 Table in S2 Appendix. The population of drug resistant strains is highly consistent with the presence of two log-normal subpopulations (S3 Fig and S7 Table in S2 Appendix). One subpopulation has a high parameter value, mean $\log_2$(MIC) = 7.4 (MIC = 55 µg/ml) with a relatively narrow subpopulation standard deviation = 0.8 $\log_2$(MIC) units. The more resistant subpopulation corresponds to approximately 71 (35%) of the 205 strains or 10% of the total 714 strains. The other strains making up approximately 65% (n = 134) of the drug-resistant population had mean and standard deviation parameters

consistent with subpopulation parameters of mean = 4.6 and standard deviation = 1.1 only slightly elevated relative to the baseline subpopulation, mean = 3.9 $\log_2$(MIC) units which was not antibiotic resistant. Thus, similar to results from *Ps. aeruginosa* strains, there appears to be a correlation between conventional antibiotic resistance and CHX resistance, although the degree of CHX resistance was low.

## A. baumannii

*A. baumannii* is another organism with increasing importance for infections because of the antibiotic resistance associated with many clinically important strains, especially strains isolated from hospitals. Prior to the 1990s, taxonomic classification of *Acinetobacter* was not uniform or well defined. The name *A. baumannii* was formally proposed and accepted in 1986 (see Peleg, et al. for a review of taxonomic and clinical features of the species [15]). As a result, there are no citations of *A. baumannii* strains isolated prior to that date. Tests of susceptibility to CHX for strains with isolation dates beginning in approximately 1999 are shown in Fig 6.

The average slope (± SEM) for the linear regression analyses was 0.11 (SEM = ± 0.0002) $\log_2$(MIC) units/year. The p-values for the F-test of linear regression model indicated extremely robust support for the linear regression fit to the data. The results indicated a mean reduction in susceptibility to CHX among *A. baumannii* strains of 2.18 $\log_2$(MIC) units or

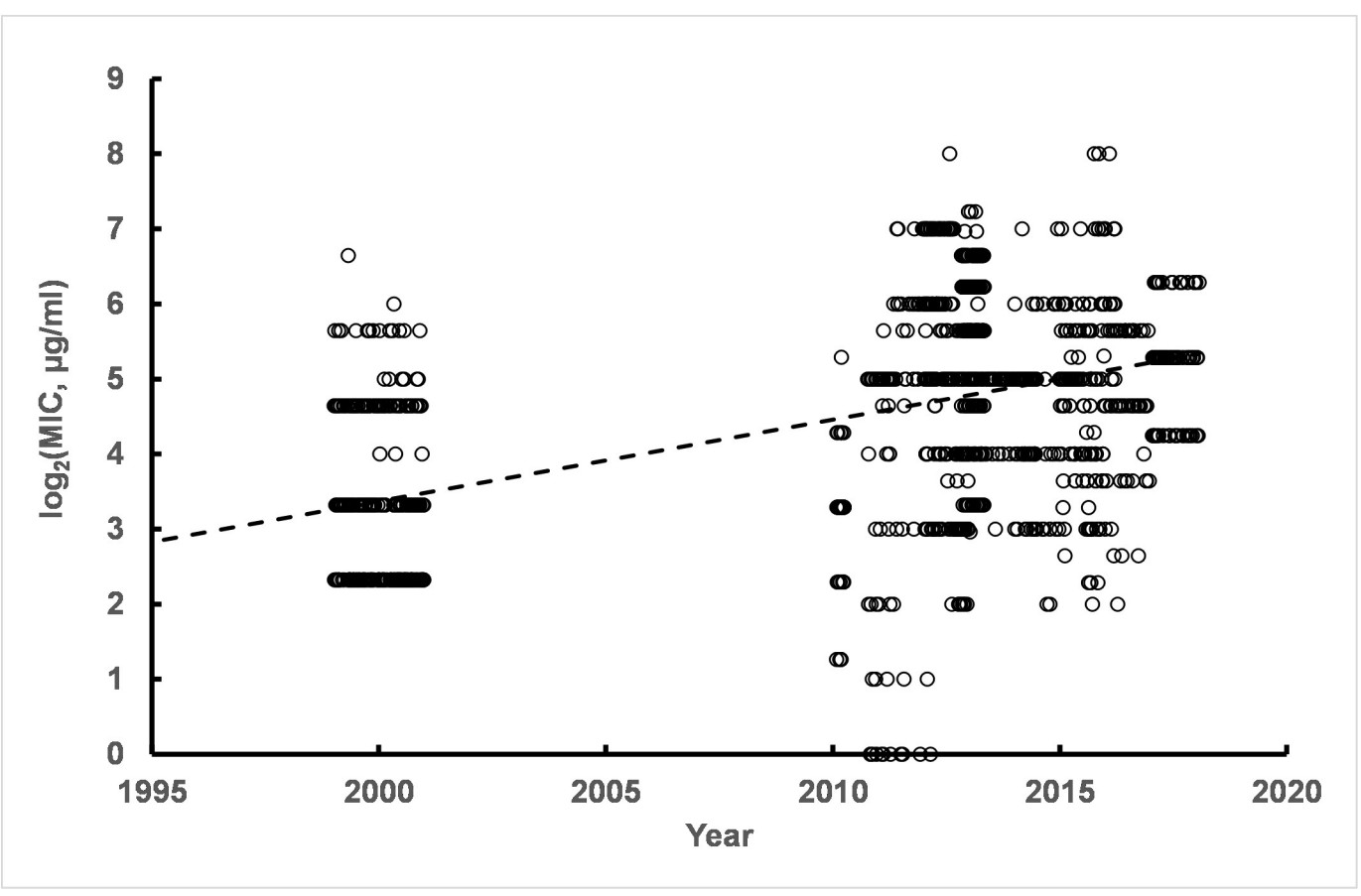

**Fig 6. Time course of MICs for *A. baumannii* for strains from all isolation dates.** Individual $\log_2$(MIC) values are shown for the 1277 readings reported in the publications compiled for this report. A representative example of linear regression from among 50 permutation runs (dashed line) is shown superimposed on the individual measurements. The parameters for linear regression and details of statistical analyses are given in detail in S8 Table in S2 Appendix.

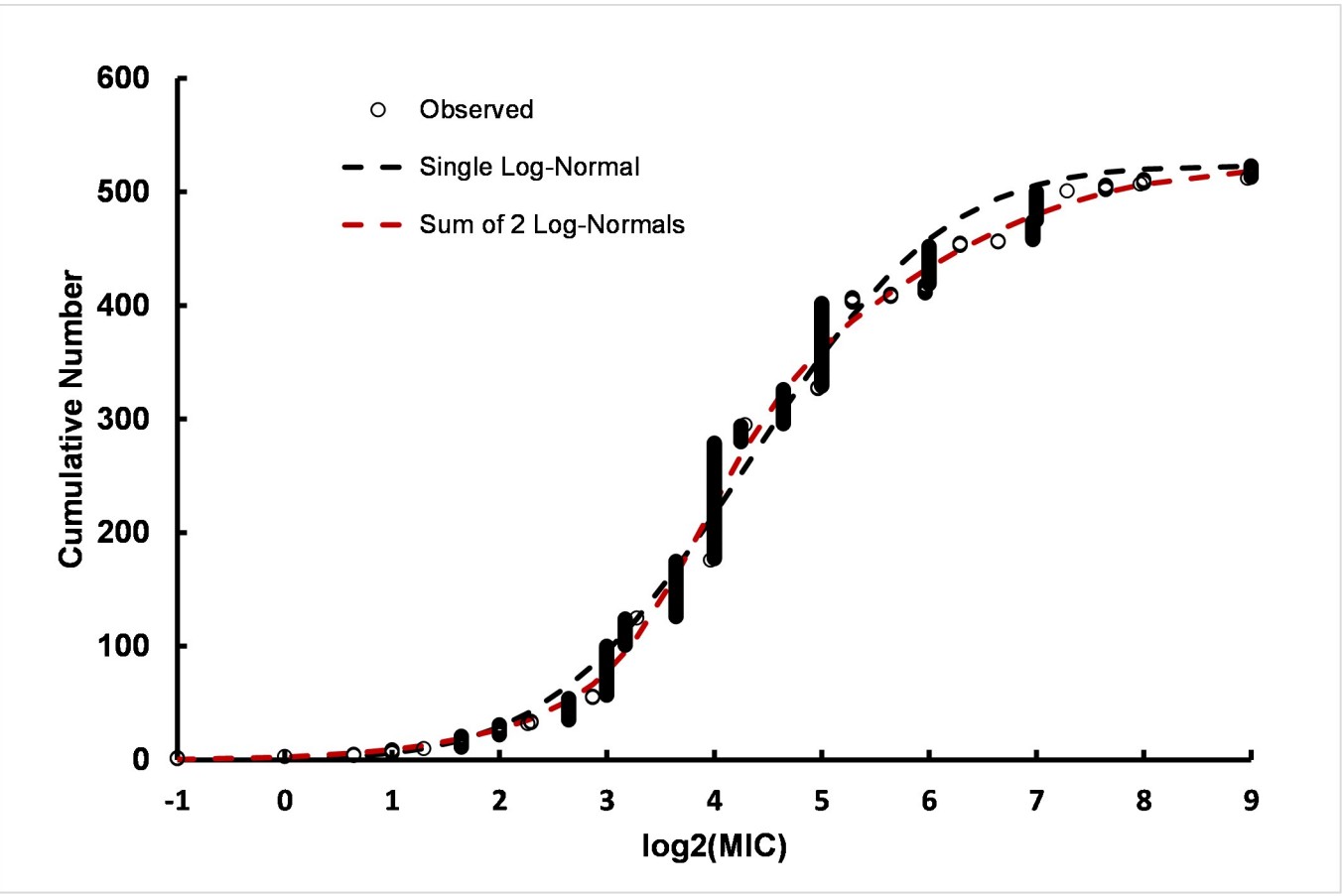

**Fig 7. Fits of a single log-normal and the sum of two log-normal distributions to MIC data from *A. baumannii* strains isolated on all dates.** Parameter values for the best fit are presented in the main text and details for the comparison of fits is given in S9 Table in S2 Appendix.

~4.5-fold, a shift from approximately 16 μg/ml in the year 2000 to approximately 72 μg/ml projected from the data for the year 2020.

Considering the density of the data in the time interval from 2010 to 2017, the corresponding $\log_2$(MIC) values (n = 983) were analyzed separately from data with earlier isolation dates (S4 Fig and S10 Table in S2 Appendix). The average slope = 0.112 (SEM = ± 0.001) $\log_2$(MIC) units. The p-values for the F-tests of the linear regression model strongly support the conclusion that the slope of the line is positive with values nearly identical to the results from the data pooled from all isolation dates (S9 Table in S2 Appendix). Thus, the resistance to CHX appears to have increased across the entire time course considered here, including over the last decade.

To further analyze the *A. baumannii* MIC data, a single or the sum of two log-normal distributions was fitted to the $\log_2$(MIC)s. The results of non-linear least squares regression show that the distribution of $\log_2$(MIC) values is substantially more consistent with a sum of two log-normal distributions than with a single log-normal distribution (Fig 7 and S11 Table in S2 Appendix). The results are most consistent with a mean of approximately 2.83 $\log_2$(MIC) units (7.1 μg/ml) CHX for one subpopulation and a second subpopulation with a mean of approximately 5.30 $\log_2$(MIC) units (39 μg/ml) CHX, that is, a 5.5-fold decrease in susceptibility to CHX. The population with the higher MIC made up approximately two-thirds (69%) of the total population. The results are substantially in agreement with the results from linear regression.

The data for MICs for strains isolated before 2010 were too sparse to analyze the sum of two log-normal distributions with confidence. However, the data are consistent with a single population with population parameters of mean = 3.2 $\log_2$(MIC) units (9.2 μg/ml) and standard deviation = 1.3 $\log_2$(MIC) units (n = 294, not shown), which is reasonably consistent with the lower sensitivity population in the MICs compiled from all dates, i.e. 7.1 μg/ml. The MICs observed using strains isolated in 2010 or later were also analyzed for the fit of a single or the sum of two log-normal distributions (S5 Fig and S11 Table in S2 Appendix). Similar to the results for linear regression of strains isolated between 2010 and 2017, fits of log-normal distributions to MICs observed after CHX treatment results were most consistent with two *A. baumannii* subpopulations, one with a subpopulation mean of 3.9 (Std Dev = 1.2) $\log_2$(MIC) units and a second subpopulation with mean of 5.8 (Std Dev = 0.8) (S5 Fig and S11 Table in S2 Appendix). The more CHX-resistant subpopulation has approximately the same population parameters as the more resistant subpopulation from the data pooled over the full time course, i.e. 1999–2017. The results are consistent with a greater concentration of CHX resistant strains after 2010.

More than 2/3s of the *A. baumannii* strains compiled with isolation dates of 2010 or later were designated by authors as multiple-drug resistant (MDR) (S12 Table in S2 Appendix). Analysis of subpopulations using fits to a single or the sum of two log-normal distributions indicate the presence of two subpopulations within the MDR strains with mean susceptibilities to CHX of 20 μg/ml and 66 μg/ml in good agreement with the values of 15 μg/ml and 55 μg/ml for all strains isolated from 2010 to 2017. This suggests that MDR strains are not necessarily also CHX-resistant.

### *E. coli*

Another Gram-negative organism which has been tested for CHX susceptibility is *E. coli*. The broad range of MIC values reported for *E. coli* strains and linear regression of $\log_2$(MIC) values over time is shown in Fig 8. The average slope (± SEM) for time versus $\log_2$(MIC) was—0.035 (± 0.00004) $\log_2$(MIC) units. Details are provided in S13 Table in S2 Appendix. The biological significance of the negative slope of the linear regression line is uncertain but suggested an increase in CHX susceptibility of 1.7 $\log_2$(MIC) units over 50 years, i.e. a change of approximately 3-fold. The negative slope was particularly associated with strains isolated from humans in the time interval after 1990 with the slope = - 0.049 $\log_2$(MIC) units associated with human strains isolated after 1990. MICs for strains isolated from animals showed a positive slope of 0.036 $\log_2$(MIC) units (not shown). However, reports of *E. coli* strains from animals were concentrated with more than 80% from two studies [16, 17], so that interpretation of results from strains isolated from animals is limited by the small number of studies compiled for this report.

Analyzing the $\log_2$(MIC) values reported for 45 antibiotic resistant strains, including carbapenem-resistant [18], extended spectrum beta-lactamase expressing cells [8], or strains expressing the New Delhi metallo-beta-lactamase [10], showed a mean (± Std Dev) of 4.14 (± 1.54) $\log_2$(MIC) units for the pooled antibiotic-resistant strains. This is different than the mean of all other strains (n = 2134) with mean = 1.81 (± 1.69) $\log_2$(MIC) units. The subpopulations were compared using Welch's t-test and results were consistent with a substantial difference in susceptibility to CHX between the known antibiotic-resistant strains and the other *E. coli* strains (S15 Table in S2 Appendix). The magnitude of the difference between known antibiotic-resistant *E. coli* and the large number of other strains was approximately 5-fold.

The fit of a single log-normal distribution was very robust (S6 Fig and S14 Table in S2 Appendix). An equation with the sum of two log-normal distributions failed to converge

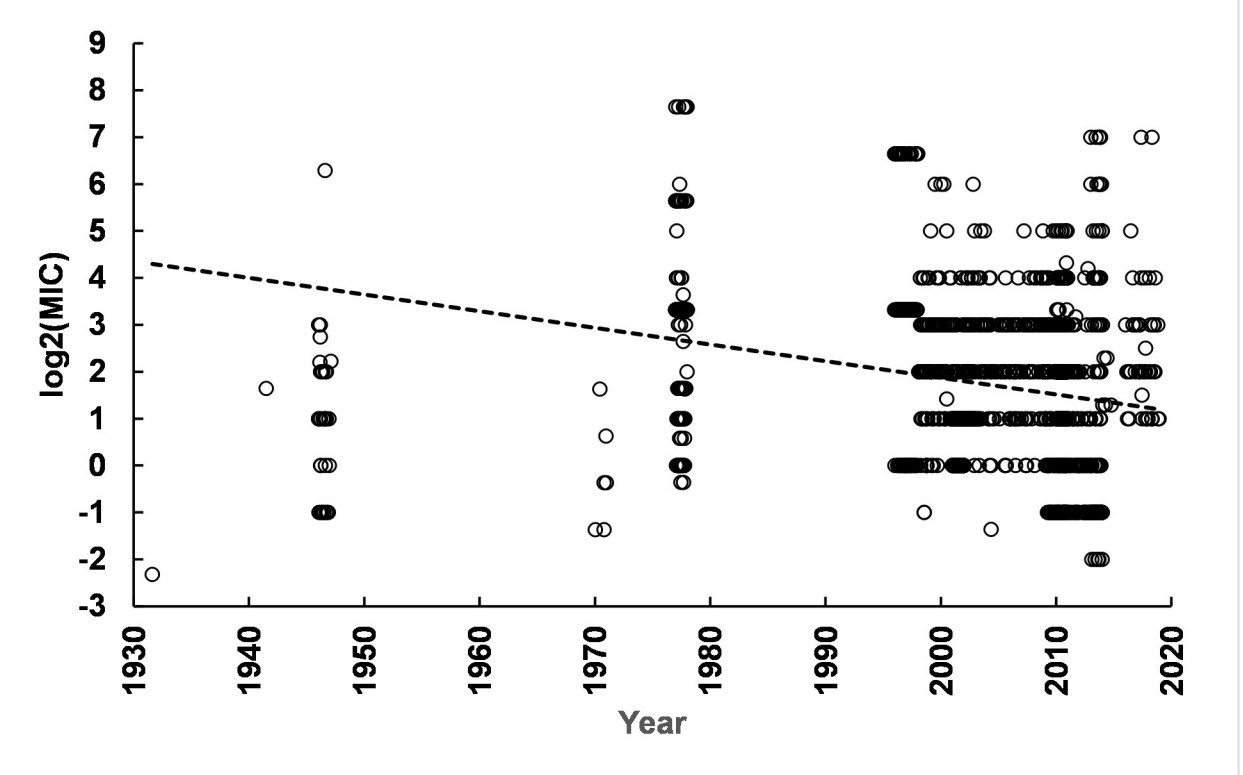

**Fig 8. Time course of *E. coli* susceptibility to CHX over time.** A large number of *E. coli* strains (n = 2179) have been tested for susceptibility to CHX with results shown for strains isolated from as early as the 1940s. Linear regression indicated a negative slope of relatively low magnitude as shown by the dashed line. Parameters are given in the text and in S13 Table in S2 Appendix. Subpopulations of *E. coli* strains with different susceptibilities to CHX were not detected (S6 Fig, S14 Table in S2 Appendix). That is, no evidence was detected of a CHX-resistant subpopulation among more than 2000 strains evaluated.

indicating that the data are modeled very well using a single log-normal distribution with no evidence of the presence of a second log-normal subpopulation. This is despite the presence of 45 values for antibiotic resistant *E. coli* strains with mean = 4. 1 $\log_2$(MIC) units (S15 Table in S2 Appendix). This may indicate that the procedure is not sufficiently sensitive to detect a subpopulation with a small number of values, i.e. ~ 2% of the total 2179 values.

## *E. faecalis*

The last Gram-negative organism included here is *E. faecalis*, another infectious agent frequently associated with antibiotic resistance, especially resistance to vancomycin. The time course of susceptibility to CHX is shown in S7 Fig and S16 Table in S2 Appendix. The slope of the linear regression line over 50+ years was negative suggesting a small decline in resistance to CHX, i.e. a reduction of -1.5 $\log_2$(MIC) units. The biological significance of the decrease is uncertain, especially since the data density is low.

The fit of log-normal distributions to CHX inhibition $\log_2$(MIC) values for *E. faecalis* were carried out for strains isolated from all dates collected for this report (S8 Fig and S17 Table in S2 Appendix). The results indicated the presence of two subpopulations with susceptibility to CHX of 5.0 µg/ml and 30 µg/ml.

The authors of reports containing *E. faecalis* data reported antibiotic resistance for 38 of the 276 MICs compiled here. The mean (± std dev) in $\log_2$(MIC) units for strains reported as VRE[+], vancomycin-resistant, or qac[+] = 2.0 (± 0.33), which is most consistent with the

subpopulation with mean = 2.33 and not consistent with the sub-population with $\log_2$(MIC) = 4.9. However, the limited number of values reported for antibiotic-resistant strains makes interpretation of the data tentative. Resolution of this possible contradiction will require more data, although it is possible that the types of antibiotic resistance found in the strains analyzed here are not related to CHX resistance.

## S. epidermidis

A Gram-positive species for which there is considerable concern regarding antibiotic resistance is *S. epidermidis*. The time course of MICs reported for CHX are shown in S9 Fig and S18 Table in S2 Appendix. The regression line had a slightly positive slope 0.005 (± 0.00004, SEM) $\log_2$(MIC) units/year. The p-values for the F-test of the fit of the linear regression model suggested no difference in susceptibility to CHX over time. Similarly, the fit of cumulative log-normal distributions to the *S. epidermidis* data indicated only for a single log-normal distribution (S10 Fig and S19 Table in S2 Appendix).

A study with analysis of a large number of *S. epidermidis* strains analyzed for susceptibility to CHX (n = 182) [19] found no relationship between prior exposure to CHX and the presence of *qacA/B*. Another study of *S. epidermidis* strains tested for CHX susceptibility [20] reported the presence of qacA/B in the strains with a reduced susceptibility to CHX. In all cases, the degree of increased resistance to CHX was low, typically a 2-fold shift. Considering the limited data available, the relationship between antibiotic resistance factors and susceptibility to CHX was not conclusive for *S. epidermidis* strains.

## S. aureus

Perhaps the most important Gram-positive organism about which there are increasing concerns of antibiotic resistance is *S. aureus*. One of the most commonly applied discriminants of *S. aureus* strains is classification based on methicillin resistance. *S. aureus* strains reported as resistant to methicillin were analyzed separately from methicillin-sensitive strains.

As shown in Fig 9, linear regression of $\log_2$(MIC) values indicated a low magnitude negative slope for data from MRSA and MSSA strains consistent with a slight increase in susceptibility to CHX over time. (S20 and S21 Tables in S2 Appendix). The magnitude of the negative slopes is equivalent for both MRSA and MSSA strains; slope = -0.036 (SEM = ±0.0003) and -0.032 (SEM = ± 0.0002) $\log_2$(MIC) units, respectively. The slopes correspond to a decrease of -1.8 to -1.6 $\log_2$(MIC) units over 50 years, respectively, i.e. a decrease of 3.4-fold for MRSA and 3-fold for MSSA.

Fitting cumulative log-normal distributions to the data indicated the presence of a single population for MSSA strains with population parameters mean = 0.23 and standard deviation = 1.33 $\log_2$(MIC) units. The fit of a sum of two log-normal distributions failed to converge on common values. No convincing evidence of a second subpopulation among MSSA strains was observed (S23 Table in S2 Appendix). The fit of a single cumulative log-normal distribution was sufficient to explain the data. Likewise, non-linear least squares regression for the fit of the sum of two log-normal distributions did not converge for the MRSA data, either (S22 Table in S2 Appendix). However, in comparison with results for MSSA strains, MRSA strains had population parameters of mean = 0.93 and standard deviation = 1.18 $\log_2$(MIC) units. Thus, MRSA strains showed a somewhat greater resistance to CHX than did MSSA strains, but both types of *S. aureus* strains showed no increase in CHX resistance over time. The difference between the MSSA and MRSA data indicated a mean of 1.2 μg/ml for CHX MICs from MSSA strains and a mean of 1.9 μg/ml for CHX MICs for MRSA strains, a 1.6-fold difference.

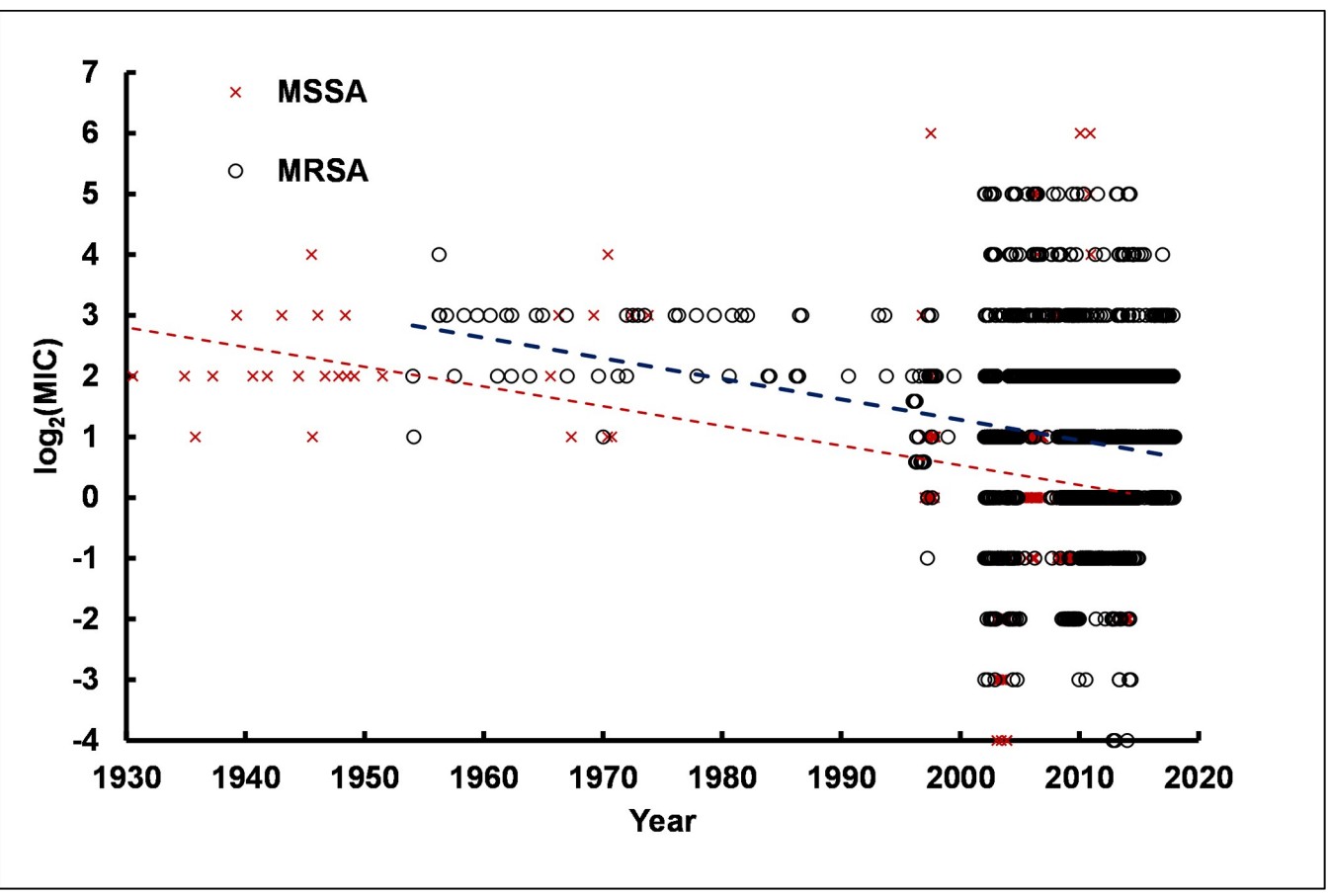

**Fig 9. Susceptibility of *S. aureus* to CHX over time.** Two sets of results are shown: Strains specified as MRSA (dark blue open circles) and strains specified as MSSA (dark red open circles). Corresponding representative linear regression lines are shown as dashed lines in the same colors as the symbols. Details of linear regression analyses are given in S20 and S21 Tables in S2 Appendix.

Statistical analysis of MRSA and MSSA differences is illustrated in S11 Fig in S2 Appendix Text showing the distribution of t-test p-values for 100 permutation runs. Analysis of the pooled MRSA and MSSA results did not detect the presence of the two subpopulations. However, this is not surprising since the usual and predominant experimental process is the use of two-fold serial dilutions. Detection of a 1.6-fold difference when the resolution of the procedure is only 2-fold would not be expected.

## *C. albicans*

A common fungal infectious agent is *C. albicans*, which has also been evaluated for increased antibiotic resistance. Susceptibility of *C. albicans* to CHX has been reported in a number of studies. A total of 1045 MICs were compiled to examine the pattern of CHX susceptibility over time. The slope of linear regression analysis over 50 permutation runs was robust with an average slope of -0.060 (SEM = ± 0.00009) $\log_2$(MIC) units. Examining the results after 1994 resulted in a similar regression coefficient = - 0.060 (not shown) indistinguishable from the estimate of the slope over the entire range of dates (S12 Fig and S24 Table in S2 Appendix). Thus, the negative slope was evident in the data collected over the last 25 years as well as the data over the entire time course, which is consistent with a small decrease in resistance to CHX over time for the pooled *C. albicans* strains.

The pooled $\log_2$(MIC) results (n = 1045) were analyzed for the fit of a single and the sum of two log-normal distributions. As shown in S13 Fig and S24 Table in S2 Appendix, there is strong evidence for the presence of two subpopulations within the strains compiled here. The sum of two log-normal distributions fit the data considerably better than did a single log-normal distribution. A more sensitive subpopulation had associated parameters of mean = 1.2 and standard deviation of 1.2 $\log_2$(MIC) units. The second subpopulation had population parameters of 3.3 and standard deviation of 0.59 $\log_2$(MIC) units. The more resistant sub-population was approximately 24% of the total 1049 strains.

The relationship between antibiotic-resistance and the subpopulations identified using log-normal distributions could not be evaluated for *C. albicans*. Only 4 strains among the 1045 for which CHX susceptibility was reported mentioned fluconazole resistance. Thus, it is possible that there is a relationship between the subpopulation with mean = 3.3 $\log_2$(MIC) units and antibiotic resistance, but the current data are not sufficient for a firm conclusion.

## Discussion

The approach used above was to consider the strains of specific microbial species as populations to be analyzed using fundamental tools of statistics. In particular, linear regression was used to determine if there was evidence of changes in susceptibility to CHX over time. A second technique applied was analysis of microbial populations to observe the presence of subpopulations with different degrees of resistance to antimicrobials within a single species. Subpopulations based on differences in resistance to antibiotics have been derived, termed epidemiological cut-off values (ECOFFs), based on criteria that are relevant in clinical medicine [21–24]. Similar methods have been applied in attempts to define ECOFFs for biocides [25]. The general concept is amenable for both antibiotics and biocides, although there are some important differences. In particular, biocides are typically used at concentrations that greatly surpass biocide resistance concentrations observed so far. Thus, in defining subpopulations of a microbial species that are resistant to biocides, the biological importance of the level of resistance observed is not clear. Nevertheless, a subpopulation with resistance greater than what was observed in a baseline subpopulation is useful for examining the potential threat of increased biocide resistance, which may also be associated with antibiotic resistance.

The relationship between resistance to antibiotics used in medicine and biocides used for disinfection in both medical and non-medical procedures has been the subject of many reports over the last 20 years (see as a few examples [3, 26, 27]). Despite considerable work, the relationship remains unclear. The work described here was intended to compile primary information in the form of MIC values for CHX reported in the literature over the last 50+ years covering as many organisms as possible. The primary question addressed evidence regarding the degree of susceptibility of various microorganisms to CHX, especially bacteria responsible for increasing numbers of human infections. An association with resistance to antibiotics commonly used in human clinical medicine was also described.

The statistical tool used previously to investigate subpopulations emphasized the use of probability density functions, e.g. the "bell-shaped curve" representative of a normal (Gaussian) distribution. Mathematically, either probability density or cumulative distribution functions can be used to describe populations. However, in contrast to the probability density form, properties of cumulative distribution functions may be more useful than probability density functions for characterization of microbe subpopulations with different antibiotic or biocide susceptibilities. This is particularly true for typical antimicrobial susceptibility testing which is based on the use of serial 2-fold dilutions. A major advantage favoring the use of cumulative probability functions is the lack of the need to place results in bins for analysis.

Although most microbiological susceptibility testing has "natural" bins based on the 2-fold serial dilutions used, the serial dilution approach itself has problems, including variable accuracy across the range of concentrations tested. Some of the problems can be avoided or their effects reduced by the use of cumulative distribution functions, as described in S1 Appendix.

Linear regression indicated an increased average resistance to CHX in some species, although the change over several decades was small. Additionally, subpopulations with different CHX susceptibilities were found in several species, including *Ps. aeruginosa*, *K. pneumoniae*, and *A. baumannii*, and these correlated to various degrees with resistance to reported antibiotic resistance. A problem in many cases was that strains tested for CHX susceptibility were not necessarily well-characterized with regard to antibiotic resistance. For example, the two-fold change in average resistance to CHX in strains of *Ps. aeruginosa* over 50 years indicated by linear regression conformed to two distinguishable subpopulations detected using non-linear least squares analysis with mean subpopulation susceptibilities of 14 μg/ml and 28 μg/ml. In *K. pneumoniae* multi-drug resistant (MDR) strains expressing EBSL and NDM-1 a subpopulation mean MIC of 164 μg/ml for CHX was detected among approximately 10% of strains compared to the baseline susceptibility of 14 μg/ml. Perhaps the most remarkable change in CHX susceptibility was detected for *A. baumannii* with a 2-fold reduction in CHX susceptibility over 20 years. CHX resistance for a subpopulation of *A. baumannii* strains showed a mean of 66 μg/ml compared to the baseline susceptibility of 7 μg/ml. However, only approximately 35% of strains designated by study authors as MDR also showed the higher resistance to CHX, i.e. 66 μg/ml, indicating that antibiotic resistance only tracked partially with CHX resistance. For these three species there were variable degrees of correlation between CHX and antibiotic resistance.

The results for *E. coli*, *E. faecalis*, *S. epidermidis*, *S. aureus*, and *C. albicans* did not indicate a general increase in resistance to CHX over time. In fact the results were most consistent with a slight increase in susceptibility to CHX over time for four of the five species. Clinical relevance of the small difference is difficult to determine. The limited data on antibiotic resistance among strains tested for CHX susceptibility among these five species rendered firm conclusions about the relationship between CHX and antibiotic resistance difficult.

Clinical relevance of the degree of CHX resistance observed in all species considered here appears limited. Park, et al. [28] mentioned that CHX concentrations used in practice are 10–40 g/L, which is many-fold higher than the MICs reported for any microbes analyzed here. Otter, et al. [29] stated that ". . .higher MICs displayed by some isolates (4–16 mg/L) are still well below the effective chlorhexidine concentration applied to skin (10000–40000 mg/L) . . ." and cited two supporting reports. McGann, et al. [30] stated "In vitro studies of MIC and MBC with CHG demonstrate bactericidal action at levels well below recommended concentrations used in practice (2000 μg/ml)." Batra, et al. [31] may have provided some of the most direct evidence for CHX efficacy in the clinic despite the resistance to CHX associated with expression of *qacA/B* in *S. aureus*. Strains carrying *qacA/B* had in vitro susceptibility to CHX 3-fold less than non-*qacA/B*-carrying strains, but infections caused by the MRSA strains were equally suppressed using a sanitation protocol at typical CHX concentrations. Otter, et al. [31] restated that high levels of CHX are used in practice: "Nevertheless, the clinical impact of such a small MBC increase is debated, since the achieved concentration of topically applied CHDN 0.1% (wt/vol) solution will yield 1000 μg/ml, many fold higher than MBC of any *qac*- or *smr*- harboring strain." It seems reasonable to conclude that the concentrations of CHX used routinely are considerably higher than those necessary to overcome the resistance to CHX reported so far. Nevertheless, the results indicate that vigilance with regard to the potential for increased resistance to CHX is warranted.

Does CHX use result in increased resistance to clinically relevant antibiotics? Wand, et al. characterized *K. pneumoniae* clinical isolates that were made resistant to CHX in vitro [32]. Five of six strains adapted to growth in CHX showed increased resistance to colistin, an antibiotic increasingly required in battling *Klebsiella* infection as an antibiotic of last resort. However, CHX resistance in the laboratory resulted from serial exposure to gradually increased concentrations of CHX, a condition apparently unlike the concentrations of CHX typically used. Are laboratory conditions used to generate CHX resistance sufficiently equivalent to conditions occurring in clinical or in food service or similar surface treatment applications to result in CHX resistance in places where the biocide is routinely used? The question remains open.

Understanding mechanisms of resistance to CHX and colistin may provide useful insights in addressing these questions. CHX potency on strains of *K. pneumoniae* isolated in the 1940s and 1950s prior to the widespread use of antibiotics [32, 33], were compared to more recently isolated strains [34]. Wand, et al. [32] and Shakib et al. [35] both reported similar patterns in expression of virulence and resistance factors with differences between pre-antibiotic and more recent strains supporting the idea that important virulence and resistance factors that were infrequently expressed in pre-antibiotic era *K. pneumoniae* strains are now commonly expressed. Although adaptive changes were reported after CHX treatment in vitro, the adaptations are not unique to CHX challenge. Surface-charge modifying mechanisms have been observed in several species of Gram-negative bacteria in response to membrane disruption challenges, including both CHX and polymyxins [36–39]. Additionally, changes in expression of a number of efflux pump systems have been observed in response to membrane disruptive agents [3, 40]. However, it is still an open question whether resistance to colistin observed after in vitro CHX exposure is clinically relevant.

*A. baumannii* is unusual in its adaptability to membrane disrupting agents, including CHX. Powers & Trent [41] summarized: "Work done over the past 5 years has shown that *Acinetobacter baumannii* has the remarkable capability to survive with inactivated production of lipid A biosynthesis and the absence of LOS in its outer membrane." Additionally, *A. baumannii* was the first organism noted to have inactivated lipid A biosynthesis in the presence of polymyxins apparently contributing to resistance. LOS-deficient *A. baumannii* strains were readily generated in vitro, but Powers and Trent concluded that isolation of such mutants in the clinic is "overwhelmingly rare" [41]. Are modifications induced by CHX treatment similar, i.e. inducible in the laboratory but very rare outside of it? Considering the remarkable resilience and adaptability of *A. baumannii* to a variety of conditions via a variety of mechanisms [42] it is not surprising that this adaptability results in some—so far indeterminate level of—resistance to CHX. *A. baumannii* clearly has particular and predominantly unique capacity for developing antimicrobial resistance. Nevertheless, resistance to CHX in *A. baumannii* appears to be rare and may be of limited clinical significance, especially when compared to concentrations of CHX typically used.

Sawyer, at al. [43] examined several LPS precursors and products within the synthetic pathway in *E. coli* when treated with a range of concentrations from 1/16 MIC to 8X MIC of a group of biocides and antibiotics, including triclosan, colistin, polymyxin B, fluroquinolones, penicillin analogues, aminoglycosides, CHX, CTAB, and the LpxC-inhibitor CHIR090. CHX treatment, in contrast to nearly every other agent tested, showed a significant accumulation of Lipid X, the 2-lipid-chain precursor of Lipid A consistent with pronounced inhibition at steps early in LPS synthesis. This mechanism appears unique compared to resistance mechanisms for other membrane disruptive agents which act via modifications of the charges on Lipid A [36, 38, 44, 45]. Thus, testing for resistant mutants with changes in early LPS synthesis seems particularly important in overall monitoring of resistance to CHX and perhaps other biocides.

Many changes in gene expression have been described after treatment with CHX. *Ps. aeruginosa* treated with 5 μg/ml chlorhexidine diacetate found 250 "statistically significant" changes in gene expression [46]. Genes with altered expression included efflux transporters and permeability-altering responses which also accompanied resistance to polymyxin B, gentamicin and EDTA. In the response of *Salmonella* to CHX a complex defense network was detected involving multiple cell targets associated with the synthesis of the cell wall, the SOS response, virulence, and a shift in cellular metabolism toward anoxic pathways [47]. Hashemi, et al. used serial exposure to generate *Ps. aeruginosa* strains resistant to 4–8 fold higher concentrations of CHX which also showed 8-32-fold increased resistance to colistin [48]. Proteomics was used to analyze the cross resistance and found that the changes occurred in proteins involved in LPS assembly in the outer membrane as well as in virulence factors. Expression of 330 of 528 proteins and colony morphology were changed consistent with changes in cell surface expression. Hashemi, at al. also stated, "Cross resistance of bacteria and colistin may be due to common features of these antimicrobials." The results are largely consistent with the results observed by Nde, et al. [46], who measured transcription changes accompanying CHX exposure in *Ps. aeruginosa*, *A. baumannii*, and *K. pneumoniae*. Upregulation of microbial metabolism, carbon metabolism, ATP binding, nucleotide binding and protein translation were observed as was downregulation of proteins involved in nucleotide binding, ATP-binding, cytoplasm and phosphate-binding proteins. Expression of many membrane-associated proteins were also changed.

Such observations are particularly important in the context of work describing mutations in core metabolic genes as part of initial events conferring antibiotic resistance preceding induction of canonical resistance factors [49]. A critical question is whether the low-level CHX resistance observed so far will be modified into relevant high-level resistance mechanisms similar to the antibiotic canonical mechanisms. The low magnitude resistance to CHX described above compiled from observations of resistance over many years may indicate frequent modifications or mutants of cell surface and metabolic processes. Such mutations and/or changes in expression of metabolic and other physiological processes may be the major effect of CHX treatment leading to low-level resistance. The changes may also result in cross-resistance to clinically relevant antibiotics. Verification or refutation of this notion may best be accomplished by examining mutations and expression changes in metabolic pathways following CHX exposure and correlating the results with resistance to CHX and antibiotics.

A workshop held in Portugal in 2012 acknowledged "that there is scientific evidence that biocides select for biocide resistance, but that there is, so far, no conclusive evidence that this also determined or will determine an increase in antibiotic resistance" [50]. A critical but unanswered question is: Is there evidence that CHX induces increased susceptibility to colistin in the clinic or community? It is clear that CHX treatment can increase resistance to colistin in laboratory conditions, but this is not the same as demonstrating that this has happened in a clinically relevant setting. Caution in interpretation and transfer of laboratory findings to clinically relevant cases has been urged over the last 15 years [51, 52], a caution worth repeating.

The work described in this report may be useful in providing evidence of the need for a national or international registry of MIC values for CHX, in particular, and biocides in general. The compiled results presented here, i.e. in S3 Appendix, may serve as a starting point for such a database by providing what may be the largest compilation of such information assembled to date. Although at this time the development of well-documented ECOFFs or similar measures for biocides are tentative, a national or international database, if supplemented with new MIC values as they are published, could provide firm bases both for ECOFFs as well as acting as a monitor for changes in biocide resistance and potential cross-resistance with widely used antibiotics.

## Supporting information

**S1 Checklist. PRISMA 2009 checklist.**
(DOC)

**S1 Appendix. Statistics for analysis of minimum inhibitory concentration (MIC) data.**
(DOCX)

**S2 Appendix. Detailed species information.** S1 Table. Linear regression parameters, *Ps. aeruginosa*, all data. S1 Fig. Distribution of p-values (F-statistic) for slope parameters, *Ps. aeruginosa* all data. S2 Table. Log-normal parameters for *Ps. aeruginosa*, all dates. S3 Table. Log-normal parameters for *Ps. aeruginosa*, pre-1990. S4 Table. Log-normal parameters for *Ps. aeruginosa*, post-1990. S2 Fig. Fit of log-normal models to *Ps. aeruginosa* data, post-1990. S5 Table. Linear regression parameters, *K. pneumoniae*, all dates. S6 Table. Log-normal parameters for *K. pneumoniae*, all dates. S3 Fig. Fit of log-normal models to *K. pneumoniae*, antibiotic resistant strains. S7 Table. Log-normal parameters for *K. pneumoniae*, antibiotic resistant strains. S8 Table. Linear regression parameters, *A. baumannii*, all dates. S9 Table. Log-normal parameters for *A. baumannii*, all dates. S4 Fig. Time-course of $\log_2$(MIC) values for *A. baumannii* strains isolated from 2010–2017. S10 Table. Linear regression parameters, *A. baumannii*, 2010 to 2017. S11 Table. Log-normal parameters for *A. baumannii* strains isolated after 2010. S5 Fig. Fit of log-normal models to *A. baumannii* strains isolated after 2010. S12 Table. Log-normal parameters for *A. baumannii* multi-drug resistant strains isolated after 2010. S13 Table. Linear regression parameters, *E. coli*, all dates. S6 Fig. Fit of log-normal model to *E. coli*, all dates. S14 Table. Log-normal parameters for *E. coli*, all dates. S15 Table. Analysis of antibiotic-resistant *E. coli* strains. S7 Fig. Time-course of $\log_2$(MIC) values for *E. faecalis* strains, all dates. S16 Table. Linear regression parameters, *E. faecalis*, all dates. S17 Table. Log-normal parameters for *E. faecalis*, all dates. S8 Fig. Fit of log-normal model to *E. faecalis*, all dates. S9 Fig. Time-course of $\log_2$(MIC) values for *S. epidermidis* strains, all dates. S18 Table. Linear regression parameters, *S. epidermidis*, all dates. S10 Fig. Fit of log-normal model to *S. epidermidis*, all dates. S19 Table. Log-normal parameters for *S. epidermidis*, all dates. S20 Table. Linear regression parameters, *S. aureus*, MSSA. S21 Table. Linear regression parameters, *S. aureus*, MRSA. S22 Table. Log-normal parameters for *S. aureus*, MRSA. S23 Table. Log-normal parameters for *S. aureus*, MSSA. S11 Fig. Distribution of p-values (t-test) *S. aureus* MRSA vs MSSA. S24 Table. Linear regression parameters, *C. albicans*, all dates. S12 Fig. Time-course of $\log_2$(MIC) values for *C. albicans* strains, all dates. S13 Fig. Fit of log-normal models to *C. albicans*, all dates. S25 Table. Log-normal parameters for *C. albicans*, all dates.
(DOCX)

**S3 Appendix. Raw data sources.**
(XLSX)

## Acknowledgments

Inspiration for this project rests significantly with two people, unfortunately now both are dead. The originator of the project was Milton Hinsch, Technical Services Director at Molnlycke. His interest as a microbiologist was to find the best information about potential resistance to CHX. The second person was a mentor to the author for about 20 years. Ferenc J. Kezdy was a consummate scientist who taught and emphasized the value of quantification in all things biological and chemical. The work reported here would not have been possible without this mentorship. A third person is also thanked as a friend and advisor over many years on topics in biostatistics, Tom Vidmar. Tom has been a teacher and an encouraging influence on

this and many other projects over many years. Of course, decisions and choices regarding analyses contained in this work are solely the responsibility of the author.

## Author Contributions

**Conceptualization:** Stephen Buxser.

**Data curation:** Stephen Buxser.

**Formal analysis:** Stephen Buxser.

**Funding acquisition:** Stephen Buxser.

**Methodology:** Stephen Buxser.

**Project administration:** Stephen Buxser.

**Software:** Stephen Buxser.

**Supervision:** Stephen Buxser.

**Validation:** Stephen Buxser.

**Visualization:** Stephen Buxser.

**Writing – original draft:** Stephen Buxser.

**Writing – review & editing:** Stephen Buxser.

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
