## [Decision Letter · Decision Letter 0]

23 Apr 2021

PONE-D-20-37954

Has resistance to chlorhexidine increased among clinically-relevant bacteria? A systematic review of time-course and subpopulation data

PLOS ONE

Dear Dr. Buxser,

Thank you for submitting your manuscript to PLOS ONE. After careful consideration, we feel that it has merit but does not fully meet PLOS ONE’s publication criteria as it currently stands. Therefore, we invite you to submit a revised version of the manuscript that addresses the points raised during the review process.

This is an  importance piece of work  and had a great value for the scientific population. Nevertheless, the paper is lengthy and should be shortened. Moreover, a major revision on the clarity of the methodology and results should be performed. 

We look forward to receiving your revised manuscript.

Kind regards,

Amal Al-Bakri

Academic Editor

PLOS ONE

2. Please confirm that you have included all items recommended in the PRISMA checklist including evaluating the risk of bias for the included studies and limitations of your study.

"This study was funded under a contract between SB and Molnlycke Health Care,

Surgical Marketing, in Peachtree, GA. The url is: https://www.molnlycke.us/contact-us/

The contact person at Molnlycke is Jason Liles. Molnlycke is the manufacturer and

marketer of products including formulations containing chlorhexidine. SB was

contracted as an independent consultant and has no other affiliation with Molnlycke or

any of its affiliates.

The funders had no role in study design, data collection and analysis, decision to

publish, or preparation of the manuscript."

We note that you received funding from a commercial source: Molnlycke Health Care.

4. Thank you for stating the following in the Competing Interests/ section:

"I have read the journal's policy and the authors of this manuscript have the following

competing interests:SB was funded as an independent contractor by Molnlycke Health

Care, USA. Independence stipulated that Molnlycke had no role in study design, data

collection and analysis, decision to publish, or preparation of the manuscript."

We note that one or more of the authors are employed by a commercial company: Select Bio Consult, LLC.

4.1. Please provide an amended Funding Statement declaring this commercial affiliation, as well as a statement regarding the Role of Funders in your study. If the funding organization did not play a role in the study design, data collection and analysis, decision to publish, or preparation of the manuscript and only provided financial support in the form of authors' salaries and/or research materials, please review your statements relating to the author contributions, and ensure you have specifically and accurately indicated the role(s) that these authors had in your study. You can update author roles in the Author Contributions section of the online submission form.

4.2. Please also provide an updated Competing Interests Statement declaring this commercial affiliation along with any other relevant declarations relating to employment, consultancy, patents, products in development, or marketed products, etc.  

Reviewers' comments:

Reviewer's Responses to Questions

**Comments to the Author**

1. Is the manuscript technically sound, and do the data support the conclusions?

Reviewer #1: Partly

Reviewer #2: Partly

2. Has the statistical analysis been performed appropriately and rigorously? 

Reviewer #1: No

Reviewer #2: Yes

3. Have the authors made all data underlying the findings in their manuscript fully available?

Reviewer #1: Yes

Reviewer #2: Yes

4. Is the manuscript presented in an intelligible fashion and written in standard English?

Reviewer #1: No

Reviewer #2: Yes

5. Review Comments to the Author

Reviewer #1: Dear Dr Buxser

Many thanks for the opportunity to review your work. This is a critically important field of work and your report has the potential to deliver important information to health professionals and policy makers.

Below I have provided some feedback which I hope that you find helpful for improving your manuscript.

Overall, I feel that the manuscript is too long. At present, I estimate that the article is >12,000 words. I fully appreciate that systematic reviews are typically longer than other manuscripts, but they rarely exceed 5000 words.

Intro

The 1st paragraph feels a little disjointed. I'm not sure lines 48-52 add important content. Could you replace these lines with (for example) the structure of the molecule and how it works against microbes, the prevalence of use in surgery / food preparation, it's major pros and cons/safety concerns, etc?

Overall, I feel the introduction would benefit from truncation, by at least half. The paragraphs concerning MICs could be summarised in to a few sentences at most - interested readers can look elsewhere if needed, or more likely, will know this information.

The final paragraph of the introduction feels more like methods - you're outlining reporting issues in studies you have included and how you plan to drawn these terms/stats together. I think this should live in the analysis part of the methods.

Methods

Please revamp the presentation of your methods, headings, etc to comply with the PRISMA guidelines - I see that you've supplied this checklist but marked many of the sections N/A. I recommend following this checklist and organising your report as such. Additionally, the Cochrane Handbook is freely available and defines how reviews should (ideally) be laid out and what content is required, etc. At present, the order is unconventional/confusing as you present your statical methods first, without describing what you're analysing in who/what/where?

The definitions could be removed as these could be considered tacit amongst readers of this work.

Lines 146-152 could be summarised into 1 sentence with a reference. If readers don't know what permutation is then they can look at your reference / search elsewhere.

Line 155 "...which were typically in the range of p << 0.00001 and often in the 156 range of 10^-10 or smaller" could be shortened to "...which were typically 10^-10 or smaller"

Lines 159-162 these looks like results? Please consider moving them

Line 217 - This is your first description of the work as "systematic" but the search has many problems & is missing important information. Overall, the search description is too verbose. The 2 searches you describe should be summarised into 1 search / redone for your revised manuscript. I recommend the NICE Health Care Database (https://hdas.nice.org.uk/) as it's user friendly, covers all the important databases and allows you to export search results in numerous formats (.bib for reference managers, word, pdf, etc). What was the lower bound of the 1st search dates (database inception, 1970, 1980, 2011)? Science Direct is not a database per se, it is merely the engine for searching other databases. At present, you're missing some important databases from your 1st search including EMBASE and clinicaltrials.gov which are considered the absolute minimum. Your 2nd search looks much more comprehensive. The supplemental searches on Google Scholar and other non-indexed databases are fine to include. The differing strategies are likely to introduce biases of selection between pre-2012 and 2013-2018 hits. How many hits did your searches find in total? How many people screened the titles & abstracts? Normally, screening of studies is performed by at least 2 independent researchers to prevent error/misses, etc - was this done? If not, I recommend that an additional author joins the team.

What software did you use for your analyses? And what code was used to regress the data? I can't see how you've handled the variance of the aggregate data.... Normally, you'd apply meta-regression techniques for aggregate data (e.g. the metafor package in R or metan/metareg packages in Stata, etc) rather than simple OLS/NLLS regression. Simple regression techniques fail to account for differences in study prevision (e.g. large studies have a small variance and thus, could/should have more weight in the model), ignore potential sources of heterogeneity, etc.

Results

Figure 1 (row 1) is incomplete

The 2nd paragraph of the results should sumamrise the included studies. Human? Adults? Time spans? Countries? Settings? Population screening vs. surgical samples? etc etc. Brevity is key

I see no value in Fig3/22

Was the risk of bias assessed for included studies? This is normal in systematic reviews and should be presented in the results too.

Each section for each microbe could be significantly shortened to improve readability.

I love the scatter plots of MIC vs year with fitted lines - however, I'd like to see these from meta-regression instead. Could you colour the dots by concentration of CHX or delivery format or source (skin, blood, nasal, etc)?

I'm sorry but I don't understand that Figs 4/5/7/8/11/12/14/16/18/20/21/24

24 figures is too many - please select the most useful few (e.g. 6) for the manuscript - the rest can be supplementary.

Discussion

I haven't commented on the discussion content as I imagine this may change substantially in a revision.

I sincerely hope these comments help you to refine this important work and I'd be delighted to review a revision.

With thanks and best wishes for 2021

Reviewer #2: Has resistance to chlorhexidine increased among clinically-relevant bacteria? A systematic review of time-course and subpopulation data

Comments for Manuscript Number PONE-D-20-37954

A meta-analysis study that looks at time-course development of resistance/changes in susceptibility to CHX by looking at reported MIC values in the literature for nosocomial microorganisms. Also, the author attempted to address the degree of susceptibility to the association with antibiotic resistance.

A very comprehensive study was conducted by the author and a very large subset of data gathered from published studies was analyzed, using various statistical approaches.

There are increasing reports on antibiotic resistance / multi-drug resistance, however, there is still little work done on disinfectant and or biocide resistance. Nor is there much reports on correlating antibiotic and biocide resistance. The mechanisms of biocide resistance remain unclear, although reports state that the mechanisms are similar to antibiotic resistance. The current paper “zooms” out and gathers numerous data on CHX resistance, and draws a time-line indicating how the MIC of CHX is increasing over time. A very relevant and necessary study, and novel indeed. I do agree with the authors on a national registry for MIC values, this will allow the monitoring of microbial resistance development.

However, the review in itself was difficult to assess, as firstly the paper is rather lengthy, and it could be shortened, and there is a disconnect between the methodology and the results.

Line 70-72, gives a timeline for measurement of CHX susceptibility and resistance for the particular study, dating back 20-50 years, however, literature searches included in the final analysis (line 228-246) are between 2013-2020, from which the data presented in this paper is compiled. This is misleading, the author should clearly state (which was later referred to), that the isolates date back in some of the studies 20-50 years, date of isolation. If there is literature stating that exposure to CHX dates back 20-50 years than this should be referred to / or the strains are all CHX susceptible. If so, this should be a search parameter in the data filtering.

The dates for the research papers assessed/included and the dates for the reporting of results also do not correlate. I might have missed this (maybe it has been reported on in the supplementary material), but this brings in an element of confusion. Line 270 reports on MIC values reported over an 80 year time interval, also depicted in Figure 2, but the methodology does not indicate the inclusion of sources for these earlier timelines, as filtering was applied to the literature searches to eventually only include papers that met the parameters set.

Gathered from the data, and I could be wrong, as I am not a statistician, it seems different statistical parameters were applied to the different subsets of data representing the different microbial populations. Seemingly creating statistical bias.

Line 197-203; (in short), the number of permutation runs, and comparison of parameter values from the number of runs, as an estimator of result stability. The number of permutation runs was 80, according to this section. However, for some microbial organisms, more permutation runs were performed, see Table 1; 100 permutations runs were performed for P. aeruginosa. It appears that the introduction of variability was not kept to a minimum.

Although mentioned, there is some disconnect between the methodology, results and discussion, regarding CHX and colistin, this is, however, extensively discussed in the discussion. However, the use of colisitin coinciding with CHX use was not listed as an inclusion criterion, and the author does not mention why this particular antibiotic is of interest.

Editorial suggestions:

Line 228-246, can be combined and this section can be shortened.

How data is presented in figures can be improved, especially in figure 19, where MSSA and MRSA S. aureus are indicated (the author did use colors), but a better presentation would be different shapes (e.g. triangles and circles). This goes for the majority of the graphs, the presentation makes it difficult to follow.

The extensive information given in the section from line 252-266, should be kept to the relevant microbial organisms discussed in the paper. Line 264 seems misleading as the author refers to data compilation extending over 20 years, but based on the inclusion and exclusion criteria for the literature searches, the published data used does not date that far back.

The author made a great effort to discuss the relationship between CHX and antibiotic resistance, in this case, colistin. The discussion can be shortened, and some general information should be given, e.g. why the relationship of CHX and colistin was looked at, etc. This type of information might not necessarily be self-evident.

Ps. aeruginosa vs P. aeruginosa? I am not aware of the use of Ps.aeruginosa, if this is correct then I accept.

6. PLOS authors have the option to publish the peer review history of their article (what does this mean?). If published, this will include your full peer review and any attached files.

Reviewer #1: No

Reviewer #2: No

---

## [Author Response · Author response to Decision Letter 0]

18 Jun 2021

PLoS ONE Response to reviewers PONE-D-20-37954

Responses to Reviewer #1:

I appreciate your time and effort in reviewing this long and complicated manuscript and will address your comments, which led to a more succinct and readable version.

The major the reason that the manuscript is long is that it covers eight organisms in detail. Many authors would split the manuscript into at least two manuscripts and possibly four or even eight. This would diminish the impact because the goal was not simply to catalogue a list of changes observed using CHX but also to show the relationships observed among the organisms. It is particularly noteworthy that subpopulations across species have changes in resistance to CHX of similar magnitude, specifically of low magnitude markedly less than the concentrations of CHX used routinely. The point is best made in presenting results from the various microorganisms in a single document. Additionally, the relationships between resistance to antibiotics used clinically and resistance to CHX have both similarities and differences among the organisms. Nevertheless, the manuscript was shortened by removing many figures and tables most of which were moved to Supplemental Materials. Since the goal was to cover as many organisms as possible based on sufficient data to draw conclusions, it was not possible to justify commenting on all of the organisms without providing the data and detailed results of analyses somewhere. 

Definitions were removed or a few were moved to Supplemental Materials, although the readership may be more broad than medical microbiologists for whom some definitions might be “tacit”. Interest in CHX and potential complications of its use extend to food scientists, supervisors of medical facilities, and many others unfamiliar with experimental microbiology and statistical techniques, especially the use of randomization to assess variance and numerical validity. Omitting all explanatory definitions would likely be a disservice to the potential broader audience. 

Overall, the manuscript was rewritten, including the introduction, methods, results, discussion and Supplemental Materials. I hope you find the introduction more focused. The Methods section was restructured completely to conform substantially to the structure suggested in the 2009 PRISMA statement for reporting systematic reviews and meta-analyses. Many details were moved to the Supplemental Materials. Comments regarding original manuscript lines 146-162 were addressed in the revisions. 

The next paragraphs address comments beginning about line 217 (prior manuscript version). 

An essential point to note is that the approach used in the manuscript in both the original and revised form is not and was not intended to be a meta-analysis. It should be noted that the rest of the title of the PRISMA statement is “… that evaluate health care interventions….” [Liberati, et al. (2009) J Clin Epidemiol 62: e1-e34]. The data presented in this systematic review are not about health care interventions, as would be applicable for a drug or other medical treatment or intervention. It is a summary of primary data based strictly on direct laboratory measurements, although there are possible practical applications in clinical medicine. A major difference between what is presented in the manuscript and meta-analysis is that summary statistics, such as odds-ratios, logistic models, incidence rates, or analyses requiring multivariate analysis within individual reports are used in meta-analysis. In the current manuscript summary statistics are derived from the raw data as complement to and in some ways the converse of meta-analysis. In using raw data the approach taken here was fundamentally different than meta-analysis and requires a fundamentally different approach for useful analysis. This was apparently not adequately pointed out in the earlier manuscript, but has now been emphasized in a number of places in the revised manuscript (e.g. lines 88-91, 204-213, 221-226). As was noted in a publication describing meta-analysis [NG Berman and RA Parker (2002) Meta-analysis: Neither quick nor easy. BMC Medical Research Methodology 2:10] “A meta-analysis also differs from a ‘pooled data’ analysis because the summary results of the previous studies, not the results on individual subjects, are combined for analysis.” This distinguishes what the manuscript contains, i.e. pooled data, in contrast to meta-analysis. Similarly, Shelby and Vaske [(2008) Understanding meta-analysis: A review of the methodological literature. Leisure Sciences 30: 96-110] described in detail that meta-analysis is about combining effect sizes from individual studies, not about combining raw data from individual studies. The statistics specified for use in meta-analysis are designed especially to adjust for the quality of summary study results, not individual measurements. The process is used to account for and minimize effects of variance by weighting summary measurements inversely with variance [see E Ahn and H Kang (2018) Korean Journal of Anesthesiology 71(2): 103-112. N Mikolajewicz and SV Komarova (2019) Meta-analytic methodology for basic research: A practical guide. Frontiers in Physiology 10: article 203]. In pooling data from multiple studies, especially with regard to characterizing effects of CHX on individual strains as a population of strains within a species, minimizing variance would reduce or minimize the ability to detect strains with properties different than the majority of strains. The goal of the data analyses provided in this manuscript was to detect and quantify strains with particular CHX susceptibilities and to quantify the proportion of altered strains compared to baseline/unaltered subpopulations. Application of statistical methods minimizing variance as is done with meta-regression and similar statistical techniques applicable to meta-analysis would hide exactly the results of greatest interest and importance. I hope that the revised manuscript makes this essential distinction clear. 

In deciding which data from which articles to include, there is a profound difference between meta-analysis and this systematic review of primary data. Screening articles to include in the meta-analysis requires judgment of whether the studies to be included were designed to address a similar enough question and contained similar enough measures of a treatment or intervention to allow combining the results into a new synthesis, the meta-analysis. In contrast, the systematic review process used here was simply to find CHX MIC values from in vitro assays. Judgement of studies to include differ broadly between the two approaches. The limited judgement required to find MIC values in a report that have been presented as numbers for individual microbial strains is much more straight-forward than trying to determine if treatment protocols and associated medical summary measurements are similar enough to combine. Meta-analysis necessarily should deal with issues of the quality of the various studies under consideration, but an assembly of raw MIC values requires no quality judgement beyond the similarity of in vitro measures of MIC determinations, which is a highly standardized experimental protocol. Importantly, a study reporting 100 CHX MICs for a particular organism will be weighted 10-fold more than a study reporting only 10 CHX MICs, which is an appropriate form of weighting for individual measurements like MICs and has the advantage of using the raw data rather than summary measures. One could construct summary measures, such as the mean and SEMs for CHX MIC values contained in a single report, but ignoring the raw data would complicate the analysis in an unnecessary and highly undesirable manner. In keeping with a systematic review uncomplicated by what is needed in meta-analysis, the raw MIC values were compiled across studies into a common pool. A second author/reviewer of the search results would add little to the process.

Comments regarding search methods and databases: 

Ideally a search would use all possible databases and broad-ranging search terms to capture every possible instance of, in this case, an MIC for CHX. Of course this is a practical and financial impossibility. Many of the databases in routine use are expensive and available predominantly to academic researchers or employees of large companies who do not individually bear the costs. Moreover, searches are most often limited to the particular subscriptions provided by a library at a particular institution. Resources are used that authors have access to and tend to favor what they are familiar with. Restricting searches by use of relatively narrow search terms, i.e. a couple forms of chlorhexidine and likely terms for MICs, and restricting searches to titles and abstracts had the desired effect of locating a large number of studies with an emphasis on collections of MICs for CHX. By an admittedly difficult to verify method, i.e. checking for missed studies using review articles, no major studies missed in the searches were found. As listed in Table S2 of Appendix A, the minimum number of MICs used in analyses in this report was 276 from 16 reports of strains isolated from approximately 1970 to 2010 (E. faecalis section of Appendix B) and the maximum was 3585 MIC values for S. aureus from 24 reports with strains isolated from the 1930s to approximately 2018 (Table A2, Appendix A and S. aureus section of Appendix B). The median number of MICs used was 930 from the median of 20 studies. How is one to judge how thorough the search results are? How is one to assess “bias”? Despite statements suggesting that searches must be optimized and bias minimized, no systematic and verifiable method for assessing bias has been put forward. There are no studies that provide anything more than highly restricted snapshots of specific searches for a few topics (see for example: Kwon, et al. (2014) Systematic Reviews 3: 135; Falagas, et al. (2008) FASEB Journal 22: 338-342; Lam, et al. (2018) Journal of the Medical Library Association 106 (2)]. There are some older studies, although not many, but these suffer because they pre-date extensive changes that have taken place in indexing and electronic citation databases in recent years, including creation and suspension of a number of databases, likely changes in search algorithms - which remain proprietary and cannot be assessed anyway, in addition to the lack of objective measures that might reveal what constitutes a useful bias-free search. The extent of coverage remains ethereal and capricious and if the extant literature comparing various databases is any indication, poorly explored. One may suggest alternative databases to search under the hope for more complete results and less “bias” but all of the terms – bias, completeness, and numerous other proposed comparative terms - remain poorly defined and poorly definable in the present context. There are no objective measures comparing databases so that one can claim bias or lack of bias while notions of either remain unfalsifiable à la Karl Popper’s descriptions. 

One recent citation of direct relevance to the present review compared Web of Science, Scopus, and Dimensions and concluded “Dimensions database has the most exhaustive journal coverage, with 82.22% more journals than Web of Science and 48.17% more journals than Scopus [Singh, et al. (2021) Scientometrics 126: 5113-5142]. Although the measurement used was total size of databases by number of journals covered, it illustrates the lack of insightful measures and the difficulties of comparing these and any other databases. The immediate relevance of the publication lies with the use of Dimensions as one of the databases mentioned in the manuscript, albeit a measurement of unverifiable reliability. If the number of studies and number of MICs compiled for microbial strains isolated over the broad range of dates is insufficient to support the conclusions based on the statistical analyses, it defies reason to think that this can be accomplished at all. Of course this is a judgement call ultimately left in the hands of the reviewer, although a population of readers might provide a broader and more complete judgment over time as well as inviting further data gathering and evaluation. Dimensions does cover ClinicalTrials.gov, although it is irrelevant to this manuscript since clinical trials do not report individual MIC values. I think you will find that EMBASE is not considered superior to MEDLINE or PubMed for basic research by many researchers and that the use of EMBASE has not supplanted MEDLINE in systematic reviews (see Kwon, et al. (2014) reference cited above). Science Direct as “not a data base per se” seems irrelevant since it does enable searching other databases, perhaps allowing a broader search than would occur using individual databases. 

The original manuscript cited the use of Excel 2013 for statistical analyses, and the point is emphasized more in the revised manuscript. As noted above, the use of meta-regression and other such software is not appropriate for the compiled raw MIC data analyzed here. 

Figure 1 has been completed as much as possible. The number of records identified before duplicates are removed was not recorded and is not informative or necessary for any obvious purpose. Why this number is among those in the PRISMA flow diagram is a bit mysterious. The number that matters is what remained after duplicates were removed. Additional records identified through other sources is also a bit ethereal. Does it include only articles used specifically for data or should it include articles included in describing and discussing key points regarding the data? Nevertheless, the number filled into the form is the number of studies found outside the searches and included in data analysis. It isn’t many, since the simple searches resulted in broad selection, including many articles that had one or no MIC values useful for the analyses. 

Figures 3 and 22 are provided – now in Supplemental Materials – to address possible questions regarding the use of p-values and heavy and appropriate criticism of their misuse. The use of the term “statistically significant” is assiduously avoided (except in one quoted passage) in keeping with ongoing efforts by statisticians to obliterate use of the term because of its consistent misuse and uninterpretability [for example, see LG Halsey (2019) The reign of the p-value is over: what alternative analyses could we employ to fill the power vacuum? Biology Letters 15: 20190174 and, especially, D Calquhoun (2017) The reproducibility of research and the misinterpretation of p-values. Royal Society Open Science 4(12); 171085]. These are merely a couple of suggestions for relevant reading among what are now hundreds of similar published discussions of how statistics has been misused and has led to a plague of what are described as irreproducible results. Instead it is much more acceptable and informative statistically to provide a collection of p-values based on randomizations, which is what these figures illustrate. Such figures may currently be of little use to non-statisticians, but the literature is increasingly clarifying the issues and solutions that need to be and are gradually and grudgingly being applied in the larger scientific community. I retained the figures albeit in Supplementary Materials but in both the original and revised manuscripts suppressed showing such results for all organisms. Some would say that the omission is an error. 

Many details have been moved to Supplementary Materials for each organism and it does read better with the brevity. 

The data values in linear regression curves are presented in both manuscript versions with CHX concentration on the ordinate as log2(MIC) values and time on the abscissa. This hasn’t changed and additional coloring based on delivery format (there isn’t any, these are in vitro measurements of CHX MICs) or source would add confusion with no additional clarity. Besides, the necessary explanation of coloring would take paragraphs added to what is already criticized for excessive length. 

The “I’m sorry but” fragment is not a sentence and I am left with guessing what the question is. Apparently, the use of non-linear least squares regression analysis to calculate parameters of species subpopulations based on CHX susceptibility was not clear in the original manuscript. The revisions were designed to clarify the meaning. Figures in the main text have been reduced to nine with the intent to emphasize properties of time course and subpopulations for the first three organisms, those with the greatest change in CHX susceptibility over time, and the last three figures to present examples of the types of alternative responses noted in three of the other five species analyzed. 

The Discussion has been shortened and its focus sharpened. 

Responses to Reviewer #2:

Thank you for your time and efforts in reviewing the long and complicated manuscript. 

Please note that the manuscript does not describe a meta-analysis. The distinction between meta-analysis and what is presented here is critical. The report is a compilation of essentially raw values from in vitro measurements of CHX treatment of eight infectious microorganisms. The distinction is more clearly pointed out in the revised manuscript. I extensively reviewed the difference between meta-analysis and a compilation of raw data values for the other reviewer and I will condense the arguments below. 

Meta-analysis relies on compiling summary measures of clinical observations, usually in the form of a risk-ratio, logistic model, incidence rate, or analysis requiring multivariate analysis. That is, a single study has a single or small collection of summary values each compiled from raw observations. Meta-analysis compiles the compiled measures/statistics into a new “synthesis” expected to summarize information across the individual studies. This is not the same as a compilation of individual values reported in various studies, which in this manuscript comprise individual values from a variety of studies all using a common and highly standardized laboratory measurement, the MIC. The distinction is essential to understanding, analyzing and interpreting what is contained in the manuscript. The revised manuscript emphasizes the distinction in several places with different contexts (e.g. lines 88-91, 204-213, 221-226). 

The main text has been shortened with many figures and tables moved to Supplemental Materials. The relationship between methods and analysis should be clear due to the many revisions in the Methods and Results sections. The relationship between searches and isolation dates has been clarified in numerous places in both sections. The searches covered results from more than the dates of the searches. That is, search results were compiled from searches that covered dates well outside of the search dates. See lines 123-132 (revised manuscript). The only filter used was to compile results using the search algorithm stated in lines 134 – 137 (revised manuscript) regardless of the date of the searches thereby covering publication dates as early as the 1940s. The earliest dates were predominantly from reference strains many of which were isolated in the 1940s and one study used strains from the Murray Collection (reference 35) including many strains isolated in the pre-antibiotic era. 

As should be clearer in the revisions, the same pattern of statistical analysis was carried out for all organisms. The pattern is use of linear regression to establish time dependence for susceptibility of strains from various isolation dates, non-linear least squares analysis to assess the presence or absence of subpopulations within the collection of strains assessed for each organism, and examination of strains declared by authors as antibiotic resistant for any relationship between that resistance and resistance to CHX. It is unclear what the comment regarding possible bias introduced by statistical analysis means. In any event, the revised manuscript should be much clearer regarding the consistent patterns of statistical analysis applied. 

Original manuscript lines 197 – 203: The number of permutation runs were not always identical, although generally 50 permutation runs were used for linear regression and 10 permutation runs for non-linear least squares regression analysis. In a few cases increased numbers of permutation runs were carried out for particularly important organisms (Pseudomonas aeruginosa) and for organisms that had less precise original data (Klebsiella pneumoniae). The primary measure of fidelity for the values resulting from analyses is the limited SEM associated with parameters regardless of number of permutation runs. This “error determination” is provided in the parameter tables found predominantly in the Supplemental Materials. To explain this in detail would add significant verbiage to the report for limited expected return in clarity in a manuscript being criticized for its length. It seems reasonable to expect that someone questioning the precision of the parameters will be able to find the details in the tables provided. 

The relationship between CHX and colistin appears because this is an idea put forth in the literature cited in the Discussion. There is significant concern that resistance to colistin is resulting from the high volume use of CHX (see references 33-35 and many not cited here). More general questions about antibiotic resistance and CHX use have been raised for more years than questions about colistin (see references 26 - 35 and many not cited here). The importance of colistin as an antibiotic of last resort is explicitly stated in lines 599-600. A review that did not include some discussion of the topics would be incomplete. Similarly, failure to discuss ideas indicating mechanisms of resistance to CHX would be incomplete, especially since small changes in resistance to CHX need to be considered in the context of altering metabolic genes which may precede greater resistance as has been observed with clinically important conventional antibiotics (reference 50 and other recent publications not cited here). 

Original lines 228-246 have been condensed and modified to conform to guidelines for meta-analysis in response to the other reviewer’s comments, despite the subject not fully conforming to the meta-analysis format. 

The goal in all figures was to use uniform symbols with minor exceptions, since the object of the exercises was to compare CHX susceptibility for a wide variety of strains isolated over many years. Part of the message is that the compiled strains are not readily distinguished, since they result from near-identical methodology, i.e. MIC determinations in a liquid dilution format. The goal was also to convey the idea of data densities, which common symbols can accomplish without excess verbiage. The number of figures and the description of results has been abbreviated with the hope that the reader will not get buried in the detail. 

Lines 252 – 266 in the original manuscript was eliminated in the revised version and the explanation applicable to line 264 appears in the comments above. Also, the reasons for discussion of colistin is included above.

Although the abbreviation for Pseudomonas aeruginosa as Ps. aeruginosa is not as common as it once was, there are thousands of examples found using Google Scholar, including more than 100 examples in publications as of May 2021. It may originate from investigators who wanted to clearly distinguish between other Gram-negative organisms, e.g. Proteus, although that is pure speculation. Nevertheless, it is unlikely to lead to confusion.

---

## [Decision Letter · Decision Letter 1]

5 Aug 2021

Has resistance to chlorhexidine increased among clinically-relevant bacteria? A systematic review of time-course and subpopulation data

PONE-D-20-37954R1

Dear Dr. Buxser, 

We’re pleased to inform you that your manuscript has been judged scientifically suitable for publication and will be formally accepted for publication once it meets all outstanding technical requirements.

Kind regards,

Amal Al-Bakri

Academic Editor

PLOS ONE

Additional Editor Comments (optional):

Reviewers' comments:

Reviewer's Responses to Questions

**Comments to the Author**

1. If the authors have adequately addressed your comments raised in a previous round of review and you feel that this manuscript is now acceptable for publication, you may indicate that here to bypass the “Comments to the Author” section, enter your conflict of interest statement in the “Confidential to Editor” section, and submit your "Accept" recommendation.

Reviewer #2: All comments have been addressed

Reviewer #3: All comments have been addressed

2. Is the manuscript technically sound, and do the data support the conclusions?

Reviewer #2: Yes

Reviewer #3: Yes

3. Has the statistical analysis been performed appropriately and rigorously? 

Reviewer #2: Yes

Reviewer #3: Yes

4. Have the authors made all data underlying the findings in their manuscript fully available?

Reviewer #2: Yes

Reviewer #3: Yes

5. Is the manuscript presented in an intelligible fashion and written in standard English?

Reviewer #2: Yes

Reviewer #3: Yes

6. Review Comments to the Author

Reviewer #2: Dear Dr. Buxser,

I want to commend you on addressing all commentaries.

To summarize:

1. As a researcher, you have identified a gap in current knowledge that requires definitive verification of whether disinfectant resistance and antibiotic resistance are linked.

2. Well-linked discussion by recalling data found in results, bolstering points being delivered in discussion sections.

3. Results are presented to allow readers to follow the flow of thought patterns and allow eventual conclusions drawn from differences in data.

4. Discussion is well written, no repetition of literature found, and literature complements statements made.

Additional:

1. - line 542 to 545 is a repeat of line 535 to 537.

2. line 590, look into the following statement. I would expect that qacA/B carrying strains would be less susceptible than non qacA/B carrying strains.

"Strains carrying qacA/B had in vitro susceptibility to CHX 3-fold greater than non-qacA/B-carrying strains, but infections caused by the MRSA strains were equally suppressed using a sanitation protocol at typical CHX concentrations "

3. Line 691, I just feel this could be more subtly put, instead of “dead” just passed. (My condolences).

Reviewer #3: (No Response)

7. PLOS authors have the option to publish the peer review history of their article (what does this mean?). If published, this will include your full peer review and any attached files.

Reviewer #2: No

Reviewer #3: No

---

## [Editor Report · Acceptance letter]

11 Aug 2021

PONE-D-20-37954R1 

Has Resistance to chlorhexidine increased among clinically-relevant bacteria?  A systematic review of time course and subpopulation data 

Dear Dr. Buxser:

I'm pleased to inform you that your manuscript has been deemed suitable for publication in PLOS ONE. Congratulations! Your manuscript is now with our production department. 

Kind regards, 

on behalf of

Dr. Amal Al-Bakri 

Academic Editor

PLOS ONE